# Mammary cell gene expression atlas links epithelial cell remodeling events to breast carcinogenesis

Kohei Saeki [1], Gregory Chang[1], Noriko Kanaya[1], Xiwei Wu[2], Jinhui Wang[2], Lauren Bernal[1], Desiree Ha [1], Susan L. Neuhausen[3] & Shiuan Chen [1✉]

The female mammary epithelium undergoes reorganization during development, pregnancy, and menopause, linking higher risk with breast cancer development. To characterize these periods of complex remodeling, here we report integrated 50 K mouse and 24 K human mammary epithelial cell atlases obtained by single-cell RNA sequencing, which covers most lifetime stages. Our results indicate a putative trajectory that originates from embryonic mammary stem cells which differentiates into three epithelial lineages (basal, luminal hormone-sensing, and luminal alveolar), presumably arising from unipotent progenitors in postnatal glands. The lineage-specific genes infer cells of origin of breast cancer using The Cancer Genome Atlas data and single-cell RNA sequencing of human breast cancer, as well as the association of gland reorganization to different breast cancer subtypes. This comprehensive mammary cell gene expression atlas (https://mouse-mammary-epithelium-integrated.cells.ucsc.edu) presents insights into the impact of the internal and external stimuli on the mammary epithelium at an advanced resolution.

[1] Department of Cancer Biology, Beckman Research Institute of City of Hope, Duarte, CA, USA. [2] Integrative Genomics Core, Beckman Research Institute of City of Hope, Duarte, CA, USA. [3] Department of Population Sciences, Beckman Research Institute of City of Hope, Duarte, CA, USA. ✉email: schen@coh.org

Although complex cellular networks are known in the mammary gland, epithelial cells are key drivers of its development and the origin of mammary carcinogenesis. Embryonic mammary stem cells (MaSCs) give rise to the epithelium's entire repertoire, which is composed of two major cell types; the tubular luminal epithelial cell sheet is enclosed by an outer layer of basal cells[1,2]. The luminal cell lineage includes hormone receptor-positive sensing cells and hormone receptor-negative alveolar cells[1,3]. Breast cancer is a heterogeneous disease on a molecular level with distinct epidemiological and phenotypic features[4], as breast cancer subtypes originate from different cell types in the mammary epithelium[3–7]. However, the relationship between the mammary epithelial hierarchy and the cells of origin of breast cancer is not totally clear, partly because the gland lineage trajectory has not been fully resolved[3,5,6].

During selective periods in a woman's life, the mammary gland increases its sensitivity to sex hormones and external stimuli, simultaneously increasing the risk for breast cancer. These windows of susceptibilities coincide with the major events during which the gland undergoes complex epithelial remodeling, including embryonic, pubertal, pregnancy, and menopausal periods[1,8]. Several single-cell RNA sequencing (scRNAseq) studies of mouse mammary gland have examined the cellular composition in the gland and its active reorganization during different windows of susceptibilities[9–14]. However, challenges, such as technical obstacles, batch effects, and lack of consensus in the lineage trajectory, prevented comparisons between different studies[15,16].

Here we tested a hypothesis that a construction of integrated data, covering key windows of susceptibilities, would comprehensively capture the reorganization of the mammary gland throughout life. Furthermore, the projection of a consensus lineage trajectory could infer cells of origin for carcinogenesis, allowing us to assess the potential link of gland reorganization to the risk of different breast cancer subtypes. As many breast cancer cases develop after menopause and exposure of estrogen or estrogen mimics are thought to promote postmenopausal breast cancer[17], we designed a new study to examine the gland after menopause and its response to external stimuli. Then, by integrating available scRNAseq data sets, we constructed a mammary cell gene expression atlas and a putative consensus lineage trajectory (https://mouse-mammary-epithelium-integrated.cells.ucsc.edu). In addition, we extrapolated the lineage-specific features to human normal and malignant breast epithelium for inferring cells of origin of breast cancer. Eventually, we appraised the reorganization of the gland over life stages to deduce their potential linkages to increased risk of specific breast cancer subtypes.

## Results

### Construction of a murine mammary cell gene expression atlas.
Four scRNAseq data sets of mouse mammary glands from the embryo, neonate, puberty, and pregnancy were identified in a public database (Fig. 1a and Supplementary Data 1)[10–13]. To address the impacts of menopause and external hormone or endocrine-disrupting chemical exposure during this period, we treated surgically menopaused mice with 17β-estradiol, progesterone, polybrominated diphenyl ether congeners (PBDEs) (i.e., environmental chemicals interacting with estrogen receptor-α[8,9,18]), or combinations of them. 17β-Estradiol treatment re-expanded the gland in the menopaused mice with increased total duct length, branching points, and terminal end bud-like structures that are considered to be active proliferation sites of the gland[9] (Supplementary Fig. 1). The addition of progesterone, together with 17β-estradiol, increased branching of the gland. Simultaneous exposure to PBDEs did tend to show weaker

regrowth of the gland (Supplementary Fig. 1). The mammary glands from these treated mice were analyzed with scRNAseq.

The five scRNAseq data sets (Giraddi et al.[10], Pal et al.[11], Bach et al.[13], Tabula Muris Consortium[13], and this study) covered eight distinct developmental/life stages (embryonic, neonatal, pubertal, virgin adult, pregnant, lactating, involuted, and menopause) across three mouse strains (C57BL/6, FVB, and Balb/c), including ~75 K total barcodes (or cells) (Supplementary Data 2)[10–13]. The data were processed to remove low-quality barcodes, potential multiplets, and non-epithelial cells (Supplementary Figs. 2–4 and Supplementary Data 2 and 3; see "Methods" for detailed descriptions). The remaining high-quality 50 K putative single mammary epithelial cells were merged using the anchor correspondence-based data integration method implemented in Seurat v3[19], which resulted in Uniform Manifold Approximation and Projection (UMAP) plots exhibiting three major clusters connected by a bridging population in the middle of its trifurcation shape (Fig. 1b, c). Inter- and intra-data set differences inherent to the experiments, mouse strains, and individuals were removed by the data integration algorithm (Fig. 1c and Supplementary Fig. 5a, b). Robustness of the data integration was confirmed by the conservation of the trifurcation-like structure, regardless of the various parameters input (Supplementary Fig. 5c).

Community detection with the Louvain algorithm identified six distinct clusters in the integrated data (Fig. 1c, Supplementary Fig. 6a, b, and Supplementary Data 4). Visualization of published marker genes[9–12] revealed that the three major leaf clusters were basal cells (Basal), luminal alveolar cells (L-Alv), and luminal hormone-sensing cells (L-Hor). Cluster (C) 2 was positive for basal cell markers such as Krt14, Acta2, Krt17, and Myl9, whereas C4 and C6 expressed a luminal epithelial marker, Krt18 (Fig. 1d and Supplementary Fig. 6c, d). There were two distinct luminal cell types: (i) L-Alv (C4), which were positive for Csn3, Lalba, Csn2, and Spp1, and associated with lactation and alveolar cells; and (ii) L-Hor (C6), which expressed Areg, Cited1, Ly6d, and Prlr, and were hormone receptor positive (Fig. 1d and Supplementary Fig. 6c, d). At the center of the trifurcation were cells from the embryonic glands (Fig. 1b–d and Supplementary Fig. 6b), which contained true MaSCs capable of differentiating into the entire repertoire of the mammary epithelium[1,10]. As UMAP dimension reduction can recapitulate lineage differentiations with well-conserved cell-to-cell continuity and global relationships[20–22], the mammary epithelium lineage trajectory was thought to initiate from the embryonic MaSCs, in the middle of the trifurcation, which differentiate into three distinct cell types. Detailed examination revealed that MaSCs at embryonic day 16 reside in the area slightly skewed to Basal clusters and start to differentiate as early as embryonic day 18 (Fig. 1d). Accordingly, the bridging clusters, C1, C3, and C5, were considered as putative progenitor clusters for Basal (C2), L-Alv (C4), and L-Hor (C6), respectively. These clusters shared the same marker genes with their differentiated counterparts (Fig. 1d and Supplementary Fig. 6c, d). They also express genes related to cellular proliferation, somatic stem cell function, mammary gland development, and/or breast cancer progression, such as Birc5, Hmgb2, and Stmn1 (Supplementary Fig. 6c, d)[23–27]. To strengthen this interpretation, CytoTRACE, a recently developed non-biased bioinformatics tool to predict the differentiation state of cells, was applied to the data[28]. The algorithm leveraged the number of expressed genes per cell to infer a lineage trajectory in the given data without prior knowledge. The results revealed that putative progenitor clusters had significantly higher CytoTRACE scores compared to the corresponding leaf clusters (Fig. 1e and Supplementary Fig. 7a). The stemness indicated by higher CytoTRACE scores gradually decreased from the cells in the embryonic glands to those in the

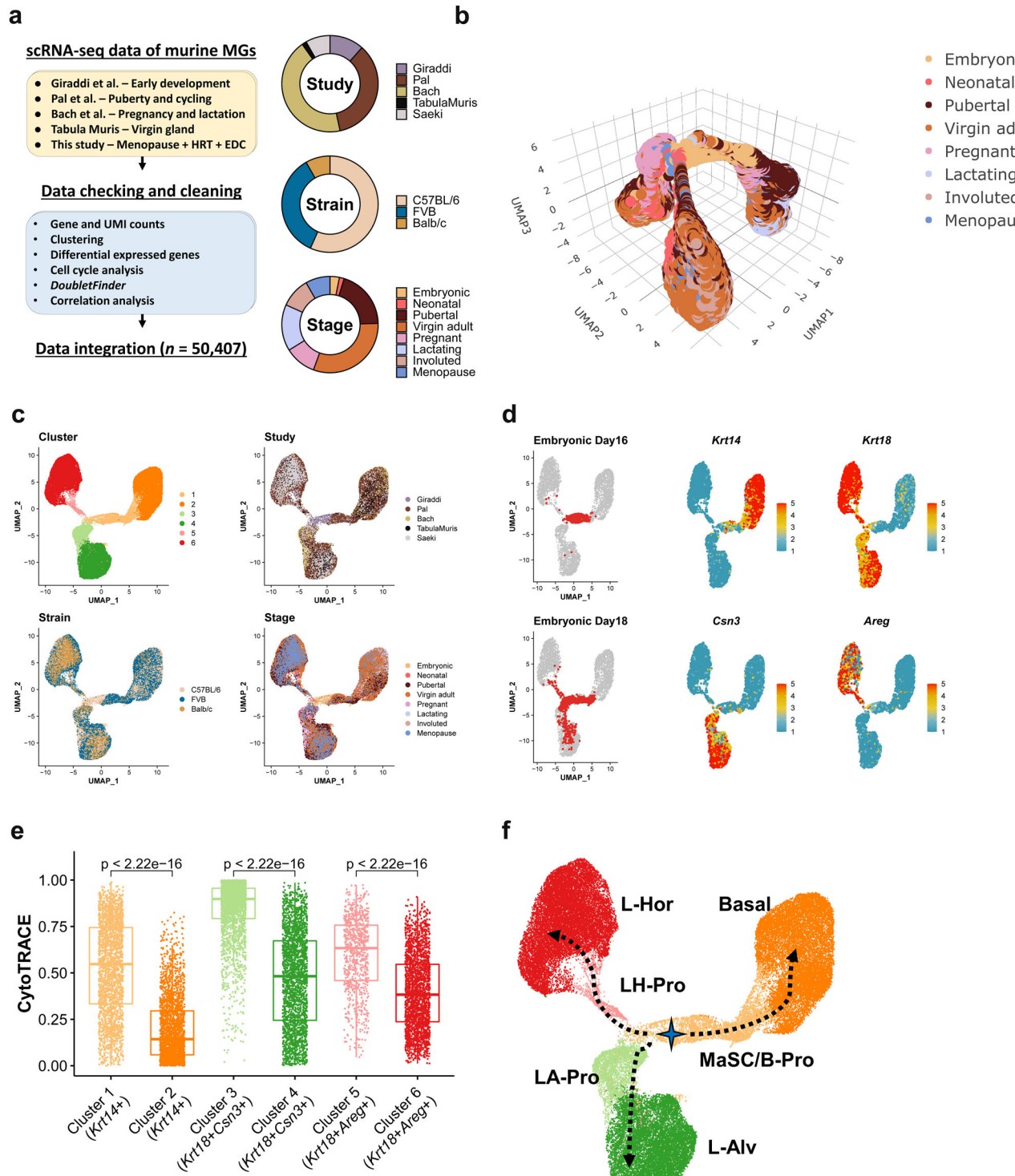

**Fig. 1 Integration of the five scRNAseq data indicated a putative lineage trajectory in the mouse mammary epithelium. a** The overview of the integration process and the data. **b** The integrated data summarized in the 3D UMAP dimensionality and color-coded by the stages at the samplings. **c** The 2D UMAP projections of the integrated data color-coded by the clusters detected with the Louvain algorithm, the original studies, the mouse strains, and the stages at the samplings. **d** The cells from the prenatal glands and the expression of the selected marker genes were visualized in the UMAP dimensionality. Five thousand cells were sampled from the entire data for UMAP visualization. **e** The CytoTRACE scores are compared between C1 ($n =$ 2404) and C2 ($n = 2393$), C3 ($n = 2659$) and C4 ($n = 2429$), and C5 ($n = 828$) and C6 ($n = 2249$) (biologically independent samples). Cliff's delta values for the comaprisons of C1 and C3, C3 and C4, and C5 and C6 were 0.72 (CI: 0.70–0.74), 0.81 (CI: 0.79–0.82), and 0.53 (CI: 0.49–0.57), respectively. CI, confidence interval. The box-plot elements were defined as follows: center line, median; box limits, upper and lower quartiles; whiskers, 1.5× interquartile range; points, outliers. **f** The inferred trajectory of the mammary epithelial starting from the embryonic MaSCs and differentiating into the three distinct differentiated states (Basal, L-Alv, and L-Hor) through the corresponding progenitor clusters.

adult glands (Supplementary Fig. 7b–d). Collectively, a mammary gland lineage trajectory was suggested to start from embryonic MaSCs (C1) and differentiate into Basal (C2), L-Alv (C4), and L-Hor (C6) clusters through the corresponding progenitor states (B-Pro (C1), LA-Pro (C3), and LH-Pro (C5), respectively) (Fig. 1f). Three additional integration algorithms (Harmony[29], LIGER[30], and scAlign[31]) yielded similar UMAP plots in overall shapes, which further supported our conclusion. (Supplementary Fig. 7e–g). The integrated data were deposited to the UCSC Cell Browser and interactively explorable on the website (https://mouse-mammary-epithelium-integrated.cells.ucsc.edu)[32].

**Trajectory reconstruction and mammary cell-type inference based on lineage-specific gene sets.** Next, the differentiation trajectory was reconstructed on pseudotime to obtain more insight into the branching timings, points of the three differentiated states, and differentiation-specific gene expression. The resulted cellular trajectory using the STREAM python pipeline[33] (see "Methods") and the marker gene expression on pseudotime supported the inferred hierarchy of the mammary epithelium in Fig. 1, showing that the embryonic MaSCs have an expression profile more similar to basal progenitors and differentiate into three epithelial cell lineages (Fig. 2a, b and Supplementary Fig. 8a, b). The putative bipotent luminal progenitor (LP) state (the S3_S0 branch) was found and predominantly occupied by embryonic cells, suggesting that luminal lineage determinations occur mainly during embryonic development. There were very few putative bipotent LPs in the postnatal glands (Supplementary Fig. 8c, d). Putative oligopotent MaSCs in the S5_S3 branch were not only composed of cells from embryonic glands[10] but also from pregnant[12] and pubertal glands[11] (Supplementary Fig. 8c). However, when the absolute number of cells was counted, they comprised only a very small fraction of the entire data set after birth (Supplementary Fig. 8d). Differences between mouse strains could not be evaluated, because cells from some important life stages are exclusively from a single strain (Supplementary Figs. 6b and 8c).

We then examined differential gene expression on pseudotime. First, we focused on the putative unipotent progenitor populations (C1, C3, and C5 clusters in Fig. 1), which were not clearly identified in the previous scRNAseq studies[10–13]. We analyzed transcriptomic changes during the differentiation process in each lineage (S0_S1: L-Hor differentiation, S0_S2: L-Alv differentiation, and S3_S4: Basal differentiation). The progenitor populations were found to express genes associated with cell cycle progression and myc pathways compared to their mature counterparts (Supplementary Fig. 8e and Supplementary Data 5).

Second, by comparing the correlation between gene expression and pseudotime, the top-ranked genes specific to each leaf state were identified (Fig. 2c, Supplementary Fig. 9a, b, and Supplementary Data 6). Using the identified genes, we built gene sets to locate selected single cells on the mammary lineage trajectory (see "Methods"). The performance of the gene sets, as measured by single-cell gene set variation analysis (scGSVA)[34], plateaued at some points during the gradual increase in the number of the top-ranked genes in the gene sets (Fig. 2c and Supplementary Fig. 9c). The best performing gene sets included the top 160, 240, 500, and 200 genes for "Stem" (S5 pseudotime), "Basal" (S4 pseudotime), "Alv" (S2 pseudotime), and "Hor" (S1 pseudotime) states, respectively (Supplementary Fig. 9c). To further evaluate the performance of the curated gene sets, we systematically examined other RNA-based features available in the Molecular Signatures Database (MSigDB $n = 22,540$; as of 3-20-2020), two outputs from CytoTRACE (CytoTRACE and Gene Count Signature (GCS)), and three basic biological characteristics

inherent to scRNAseq barcodes (gene count, transcript count, and percentage of the mitochondrial genes) for their potential association with specific states. As a result, the curated "Stem" gene set outperformed all the other features or algorithms regarding correlation with S5 pseudotime (Fig. 2c and Supplementary Data 7). The appearance of GCS, CytoTRACE, and gene sets associated with cellular stemness and proliferation at the top of the list supported proper estimation of the lineage trajectory. The other three curated gene sets also showed the best correlation with the corresponding pseudotime (Supplementary Fig. 10).

Identification of the lineage-specific gene sets motivated us to pursue de novo reconstruction of the mammary epithelium trajectory for a more flexible and less computationally intensive integration of scRNAseq data, as reported to estimate the differentiation status of human oligodendroglioma cells[35]. We summarized the scRNAseq data sets using the scGSVA scores for the curated gene sets and the UMAP dimensionality (Fig. 2d). The obtained cell distribution recapitulated analytical results of the integrated data (Figs. 1c and 2d). Combining the data from three gene sets for the differentiated states ("Basal," "Alv," and "Hor") also plotted cells on a similar trajectory (Supplementary Fig. 11a). The scores categorized cells from individual data sets into three distinct differentiation states on ternary plots (Supplementary Fig. 11b). Quantitative evaluation of each cluster's scores matched the expectation that "Stem" score is higher in cells from the putative progenitor clusters, whereas the other three lineages' scores were higher in their corresponding clusters (Supplementary Fig. 11c).

**Lineage dissection of the human mammary scRNAseq data.** The biological similarity between human and mouse mammary glands has been well documented[3,36]. To evaluate the conservation at a single-cell resolution, we unbiasedly transferred cluster annotation of the mouse integrated data to publicly available human scRNAseq data through canonical component analysis-based anchor identification (see "Methods")[19,37]. The three major lineages identified in the mouse mammary epithelium (Basal, L-Alv, and L-Hor) were found to largely correspond to B/Myo, L1.1/L1.2, and L2 clusters, respectively (Fig. 3a, b and Supplementary Fig. 12). The results supported that there is a close similarity in the biology of mouse and human mammary epithelial cells. Although the label transfer was able to meaningfully annotate the human breast epithelium, the agreement differed between individuals and inconclusive annotations were observed (Fig. 3a, b and Supplementary Fig. 12). Thus, lineage-specific gene sets could be optimized for each species, despite their shared basic biology.

The scRNAseq data from four individuals were integrated, a putative lineage trajectory was generated with the STREAM pipeline, and lineage-specific gene sets were identified as performed in the integrated mouse mammary gland data (Fig. 3c, Supplementary Fig. 13, and Supplementary Data 8 and 9). One noted limitation was that data from four individual adults probably did not include enough stem and progenitor cells. Although both UMAP and STREAM suggested a junction point of the three lineages in human breast epithelium, this point might not be the true origin point based on the inconclusive CytoTRACE scores and the transferred annotation in which the MaSC/B-pro cells were clustered in the middle of basal cells (or B/Myo cluster) (Supplementary Fig. 13b). Consequently, we decided to adopt the stem gene set identified in mice to infer a trajectory of the human mammary epithelium, assuming that they share a similar, although not identical, differentiation machinery.

Breast epithelial cells (24 K) from four adults were scored on a single-cell basis using the curated gene sets and were summarized by UMAP dimension reduction (Fig. 3c). As expected, the three

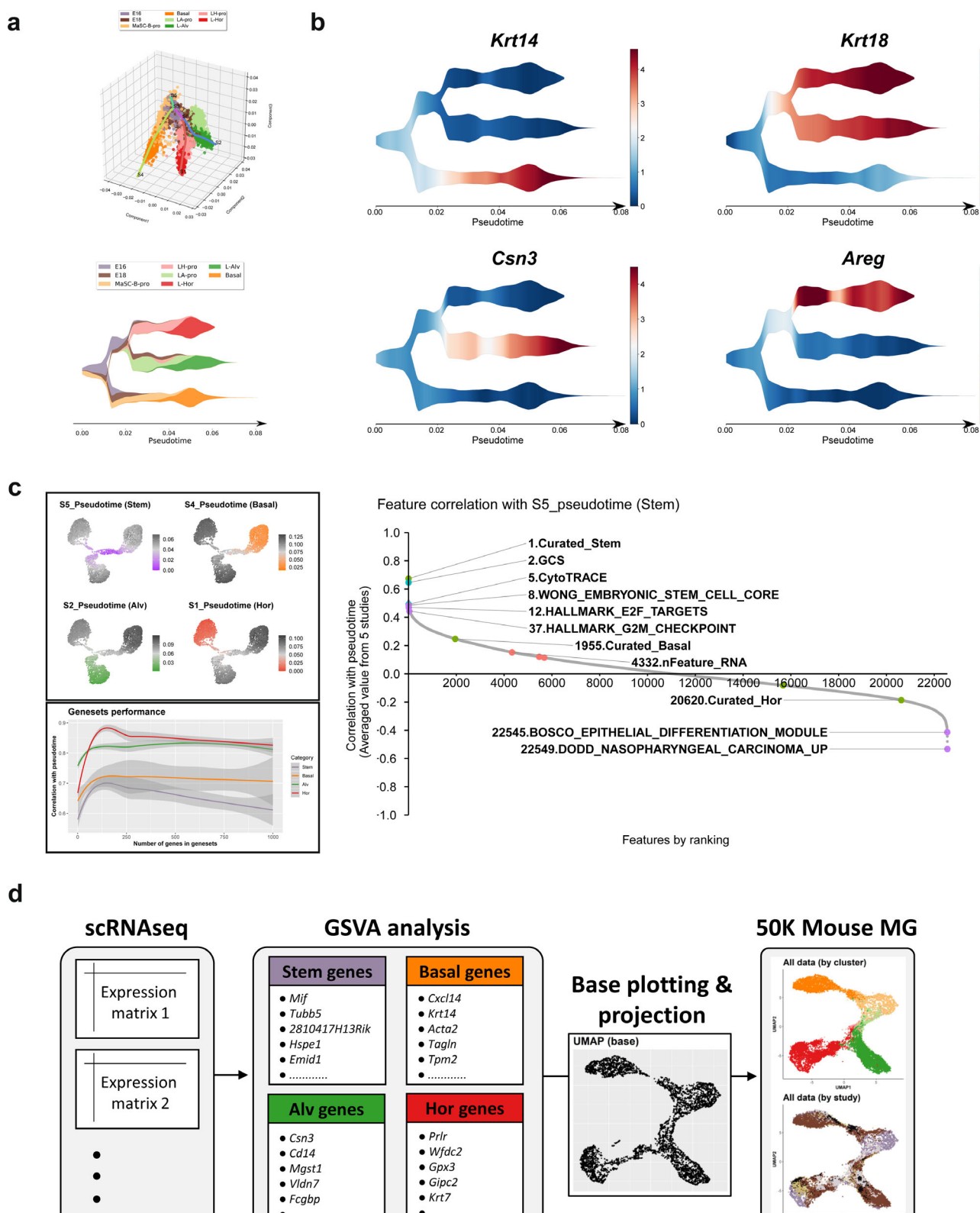

major clusters were separated and there was no bridging cluster between them, indicating the absence of true MaSCs in the human adult breast. B/Myo, L1.2/L1.2, and L2 cells showed higher scores for Basal, Alv, and Hor gene sets, respectively (Supplementary Fig. 14a–c). The GSVA scores for the stem gene set showed a significant correlation with unbiasedly calculated CytoTRACE scores (Supplementary Fig. 14d). Besides, MaSC/B-

pro cells in transferred annotation clustered together, budding from Basal cells on the UMAP plot (Supplementary Fig. 14b). These observations suggest that the adopted stem gene set from mouse can be used to predict a lineage trajectory in the human breast. Also, as observed in the analysis of mouse data, ternary plots based on human Basal, Alv, and Hor gene sets could separate the three differentiation states in individual samples

**Fig. 2 The trajectory inference of the mouse mammary epithelium scRNAseq data based on the lineage-specific gene sets. a** The lineage trajectory along pseudotime was learned in the MIle space using the *stream* pipeline and visualized on a stream plot color-coded by the combination of the stage and clusters. **b** The marker gene expressions are visualized on the stream plots along pseudotime. **c** Identification of the lineage-specific gene sets that outperform the existing RNA-based features, algorithms, and cell characteristics. The pseudotime starting from the four distinct leaf states ("Stem," "Basal," "Alv," and "Hor") are visualized on the UMAP dimensionality. Five thousand cells were sampled from the entire data for UMAP visualization. Then, top-ranked genes were identified for each pseudotime and performance of the gene sets were evaluated regarding correlation to the corresponding pseudotime with the gradual increase in the number of the top-ranked genes included in the gene sets. The X-axis represents the number of genes in the gene sets and the Y-axis represents the correlation coefficients of the scGSVA scores for the gene sets to the corresponding pseudotime. The plotted lines indicate LOESS regression of the results from the five studies and the colored area represents the confidence intervals. Finally, the performance of the curated gene sets were compared to other RNA-based features. Correlation of the scGSVA scores for the examined features to S5 pseudotime ("Stem" state) is shown as an example. The X-axis represents the rank of the features in terms of the correlation coefficients and the Y-axis represents the correlation coefficients of the scGSVA scores with the S5 pseudotime. The results for the curated gene sets, the algorithms, the selected RNA-based features, and the cellular characteristics are colored in green, blue, purple, and red, respectively. The numbers on the text labels indicate the ranking. **d** The workflow of the lineage trajectory inference and the data integration (5 studies, $N = 50,407$) based on the mouse lineage-specific gene sets with the scGSVA and the UMAP dimensionality.

(Supplementary Fig. 15a). To validate the robustness of the obtained gene sets, scRNAseq data of human breast epithelium from another three individuals[37] were analyzed. The data were processed according to the original publication and evaluated by the curated gene sets (Supplementary Fig. 15b, c). The results showed that the curated gene sets could indicate lineages of cells from another data set generated using a different modality. When mouse and human lineage-specific gene sets were compared, both commonalities and differences were recognized (Supplementary Fig. 15d). Although the known lineage markers and the relevant gene signatures were preserved in the two species, many genes were species specific (Supplementary Data 10 and 11).

**Putative cells of origin for breast cancer inferred by human lineage-specific gene sets.** Cancer is thought of as the clonal expansion of a single transformed cell[6,38]. As such, bulk transcriptome analysis of the tumor tissue could reflect features of cells of origin. Therefore, lineage inference based on the curated gene sets would predict their cells of origin as previously attempted[7,37]. Here, the pseudo-RNAseq data by combining luminal and stromal cells from the scRNAseq data confirmed that the lineage inference by the curated gene sets is robust, irrespective of sampling size and stromal contamination (Supplementary Fig. 16a–d; see "Methods"). The gene sets could also successfully infer the cell lineages of the real microarray data of the fluorescence-activated cell (FACS)-sorted population from both mouse and human[7,36] (Supplementary Fig. 16e). Confirming the applicability of the bulk transcriptome analysis method, breast cancer RNAseq data in The Cancer Genome Atlas (TCGA, https://www.cancer.gov/tcga) was retrieved[39], which included 1102 primary breast cancer and 113 normal breast tissues. The breakdown by PAM50 subtypes were as follows: Basal-like (Basal; $n = 194$), Her2-enriched (Her2; $n = 82$), Luminal A (LumA; $n = 567$), Luminal B (LumB; $n = 207$), and Normal-like ($n = 40$). The scGSVA scores for the curated gene sets were scored per sample and results were visualized by UMAP dimensionality (Fig. 3d). Three distinct clusters were identified: one almost exclusively composed of Basal tumors, one enriched for normal mammary glands, and another included the Her2, LumA, and LumB subtypes. This lineage gene set-based classification showed clearer cluster separation compared to the summarized data using the whole transcriptome (Supplementary Fig. 17a). Detailed investigation of each score revealed that breast cancer tissues had higher Stem scores compared to normal mammary tissues (Supplementary Fig. 17b). Normal-like tumors were scattered on the plot and showed a similar profile to normal mammary tissues, except for the higher Stem scores. Basal tumors showed lower scores for the Hor gene set and higher scores for the Alv gene set. On the

other hand, Her2, LumA, and LumB tumors had higher scores for the Hor gene set, with lower Stem scores for LumA. Contamination of stromal cells had little influence on the scores observed (Supplementary Fig. 17c).

When the transcriptome of human breast cancer was assessed in more detail using the putative progenitor cluster-specific gene sets defined in the mouse epithelial cell data (Supplementary Data 4), LumB and Her2 subtypes had higher LH-pro scores when compared to LumA subtype (Supplementary Fig. 17d). In contrast, LumA tumors had higher L-Hor scores. A simplified visualization with the three differentiation-associated gene sets on a ternary plot supported the idea that most of Basal tumors originated from the alveolar lineage, and Her2, LumA, and LumB tumors had their origins in the hormone-sensing cell lineage (Fig. 3d). The normal mammary tissues located at the center of the ternary plot reflected the nature of the tissue composed of all three lineages (Supplementary Fig. 16a, c). It was notable that a small fraction of Basal and Normal-like tumors were biased toward the basal cell lineage. Additional examinations revealed that they were histologically classified as metaplastic carcinoma and molecularly as claudin-low breast cancer (Supplementary Fig. 18a, b).

Triple-negative breast cancer (TNBC), a clinical phenotype, is classified into six (TNBCtype), or more recently four (TNBCtype-4), subtypes by their molecular signatures[40,41]. When TNBCs in the TCGA BRCA data sets were evaluated in light of the differentiation lineage, most TNBC tumors were mapped onto the Alv lineage (Supplementary Fig. 18c). However, the luminal androgen receptor (LAR) subtype TNBCs were scattered into the Hor lineage, indicating their different origins in the gland hierarchy. The *BRCA* gene mutation status contributes to another dimension of heterogeneity in breast cancer. It has been reported that the majority of *BRCA1* tumors are basal-like and *BRCA2* tumors are mainly LumB[42]. In accordance with the subtype-lineage relationship in Fig. 3b, *BRCA1* tumors were found in the Alv area, whereas *BRCA2* tumors were observed in both the Alv and Hor lineage with higher Hor scores (Supplementary Fig. 18d). Although age at diagnosis has also been associated with intrinsic subtypes, there was no correlation between age and lineage scores in this cohort (Supplementary Fig. 18e).

In addition, the gene set-based lineage inference was applied to scRNAseq data of human breast cancer from Chung et al.[43]. The data set consisted of 317 tumor cells from 10 patients (Basal; $n = 1$, Her2; $n = 2$, LumA; $n = 1$, LumB; $n = 4$, and Normal-like; $n = 2$). The summarization with the curated gene sets showed that Basal and Normal-like tumor cells had distinct lineage features from others (Fig. 3e), whereas tumor cells were separated mainly by individuals when a whole transcriptome was considered (Supplementary Fig. 19a). Visualization on a ternary plot

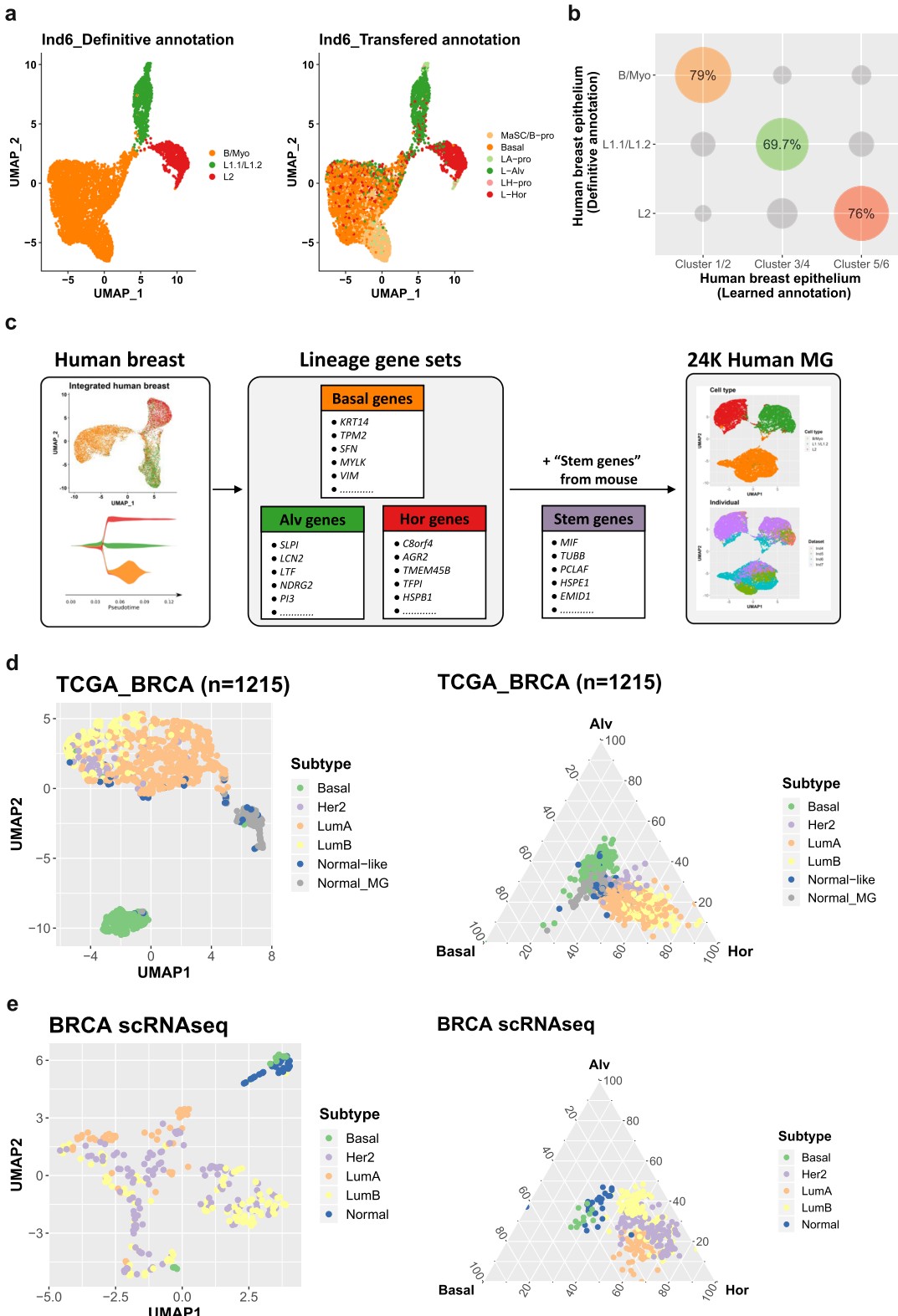

indicated the similar cellular origin as indicated in the TCGA data; Basal tumors were biased to the alveolar lineage, and Her2, LumA, and LumB tumors were located in the area for the hormone-sensing cell lineage (Fig. 3e and Supplementary Fig. 19b). Two Normal-like tumors in this study seemed primed towards the alveolar lineage. Cancer cells from each patient clustered together on the ternary plot, which also supports the clonal nature of

cancer deriving from a single transformed cell with its specific location on the lineage trajectory (Supplementary Fig. 19c).

**Mammary gland reorganization throughout life and its implication for the risk of specific breast cancer subtypes.** Lastly and inversely, we tried to classify the constructed

**Fig. 3 Lineage dissection of the human mammary scRNAseq data and the cells of origin of breast cancer. a** The UMAP projection of the human normal breast epithelial (Individual #4) clustered and annotated based on the definitive marker gene expressions or the label transfer using the integrated mouse data as a reference. **b** The agreement between the definitive annotation and the label transfer for the human normal breast epithelium from the four individuals. The progenitor and differentiated clusters from the same lineage in the mouse were merged and the cells are partitioned by the definitive and transferred annotation. The point size indicates the percentage and the sum of each column equals to 100. **c** The workflow of the lineage trajectory inference and the data integration (4 individuals, $N = 24,377$) based on the human lineage-specific gene sets with the scGSVA and the UMAP dimensionality. The "Stem" gene set was adopted from the mouse data. **d, e** The UMAP projection and the ternary plot based on the four ("Stem," "Basal," "Alv," and "Hor") and three ("Basal," "Alv," and "Hor") gene sets, respectively, for the TCGA breast cancer RNAseq data (**d**) and the human breast cancer scRNAseq data (**e**).

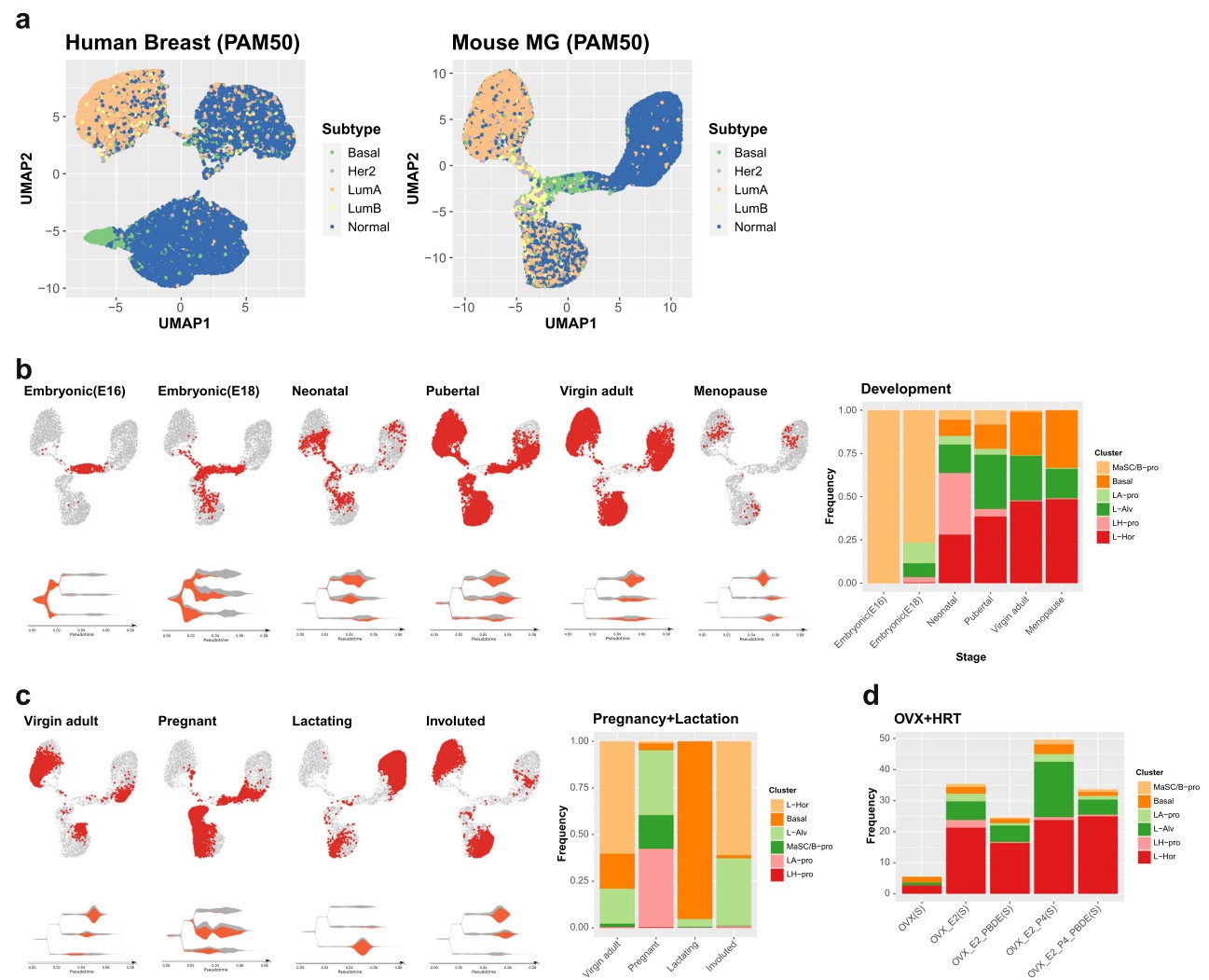

**Fig. 4 The dynamic reorganization of the mammary gland through the development, pregnancy, and HRT, and its implication for the risk of specific breast cancer subtypes. a** PAM50 molecular subtyping classification at single-cell resolution in the human and mouse mammary gland visualized on the UMAP dimensionality. **b, c** The mammary gland development and reorganization through the different life stages (**b**) and pregnancy (**c**). The cells from each stage are plotted in red on the UMAP dimensionality and stream plots. Five thousand cells were sampled from the entire data for background UMAP visualization. The proportion of each cluster in the mammary epithelium is summarized in the stacked bar plot. **d** The mammary gland reorganization during menopausal HRT and exposure to the PBDEs. The changes in proportion of each cluster in the entire gland are summarized.

mammary gene expression atlas using the features associated with breast cancer intrinsic subtyping. The PAM50 molecular classification at a single-cell resolution revealed that expression patterns of genes that infer the breast cancer intrinsic subtypes differ between epithelial cell types, which resulted in the varying subtype inference for each cluster (Fig. 4a and Supplementary Fig. 20). The similarities in the corresponding clusters between mouse and human (Basal and B/Myo; L-Alv and L1.1/L1.2; L-Hor and L2) supported the conserved biology in both species

(Supplementary Fig. 21a). Cells with similar gene expression to Basal tumors were preferentially found in the progenitor population from basal and L-Alv cells. Her2 and LumB tumor-like cells accumulated in the LP population, whereas the L-Hor cluster were a major source for LumA-like cells.

We further investigated the reorganization in the mammary gland under different intrinsic and extrinsic stimuli in light of the lineage trajectory and inference for the specific types of breast cancer. MaSCs found in the embryonic glands started to

differentiate as early as embryonic day 18 and the neonatal gland had a full repertoire of the differentiated cell types (Fig. 4b). In the fully developed glands with the influence of ovarian hormones, many cells had LumA-inference other than Normal-like subtype (Supplementary Fig. 21b). During pregnancy, the gland had the highest CytoTRACE scores (Supplementary Fig. 7b) and went through intensive reorganization with increased basal and alveolar lineages, and their respective progenitors, which potentially increased the risk of Basal tumor development (Fig. 4c and Supplementary Fig. 21c). In contrast, the lactation period was characterized by the dominance of mature basal and alveolar cells, and lowest CytoTRACE scores. After involution, the gland was repopulated by the three types of differentiated cells.

After menopause, with very low endogenous hormone levels, the mammary tissue is thought to be hypersensitive to the exposure of estrogen or its mimics (such as PBDEs)[9,17]. At the single-cell resolution, 17β-estradiol induced regrowth of the hormone-sensing lineage and the addition of progesterone further induced expansion of the alveolar and basal lineage in the gland, both of which led mainly to an increase of LumA cells in PAM50 annotation (Fig. 4d and Supplementary Fig. 21d, e). PBDEs counteracted the hormones' effects by reducing the total proportion of epithelial cells in the gland. In the murine estrus cycle, the gland is under the strong influence of progesterone at the diestrus phase. The alveolar and basal lineage progenitor populations increased at diestrus, consistent with changes during pregnancy and treatment with progesterone in our model. (Supplementary Fig. 21f). The fluctuation in the well-known secondary soluble messengers from the L-Hor lineage [*Areg* (Amphiregulin) and *Tnfsf11* (receptor activator of nuclear factor-κB ligand, RANKL)], responsive to estrogen and progesterone, respectively, reflected the stage- and experiment-dependent influences of these hormones (Supplementary Fig. 21g).

## Discussion

Technical advancement on scRNAseq analysis of the mammary epithelium has expanded our understanding of its biology, which had mainly been investigated by population-level analyses through isolation of distinct, individual cell types. However, the lack of a consensus lineage and the inherent differences between scRNAseq studies have limited interpretations of the individually collected data sets. Thanks to improved analytical tools for scRNAseq data, our current study, which included five independent data sets across three mouse strains, with four different integration algorithms, revealed a putative lineage trajectory that covered most key developmental stages of the mammary gland. The integrated mammary gene expression atlas and its reflection to cancer transcriptome support the previously suggested differentiation trajectory and cells of origins for human breast cancer. Our analysis also identified the putative unipotent progenitor populations, adding important clues towards understanding adult gland homeostasis and breast carcinogenesis. Finally, by referring the scRNAseq data to the lineage trajectory and inferred cells of origin, we visualized how different developmental stages and external hormonal exposures can alter the cellular makeup of the mammary epithelium, and ultimately evaluated the gland's risk of developing specific breast cancer subtypes.

Our pseudotemporal reconstruction of the integrated data indicated that the differentiation from MaSC into the three epithelial lineages occurs only in the embryonic gland, indicating that the three lineages would be maintained by the unipotent progenitors in the adult gland. Putative MaSCs could be present in postnatal glands, but their multipotency would be restricted in normal physiological conditions as indicated in a recent study[44]. These results were consistent with the emerging concept of

mammary gland development as revealed by lineage tracing studies[1,45,46] and scRNAseq analyses[2,10,37]. For clarification, different names have been given to the same cell types in the mammary gland. L-Hor cells are analogous to hormone receptor (HR) positive mature luminal cells (ML) and L-Alv cells correspond to estrogen receptor (ER) negative LPs, or secretory alveolar progenitors, which expand in response to progesterone, during pregnancy, and the diestrus phase[3]. There were scRNAseq studies on the adult mouse mammary gland that reported the presence of intermediate cell types between L-Alv and hormone-sensing lineages, which potentially inferred the presence of bipotent LPs in adult glands[9,11,12]. However, our analyses suggested that the suggested clusters were composed of multiplets of the cells from the two luminal clusters. Furthermore, a luminal intermediate cluster was not found in the other scRNAseq studies[10,13,14,37]. The lineage tracing studies have also revealed that L-Alv and L-Hor clusters are sustained by the unipotent progenitors in the adult gland[3]. The results of a recent scATACseq study support this interpretation but also suggest potential plasticity within the mammary gland[47]. Therefore, physical validation of bipotent LP cells in fetal and adult glands will be critical for a definitive conclusion. By filtering and integrating the multiple data sets, the trajectory from the current study could separate the putative unipotent progenitor populations, which have not yet been identified in prior scRNAseq studies[9–13]. The indicated characteristics of the progenitor cells, such as proliferative capacity and myc pathway activation, have been repeatedly associated with progenitor populations in other tissues[48,49]. Although further validation studies are warranted, the obtained gene and pathway lists could be useful resources to explore the repopulation and differentiation machinery in each lineage of the mouse mammary gland.

In our analysis, the curated gene sets guided by the lineage pseudotime could locate mammary epithelial cells on the trajectory and outperform the existing molecular profilers for both mouse and human cells. Furthermore, transfer of the murine "Stem" gene set to human scRNAseq data inferred that the three distinct lineages in human adult breast epithelium were maintained by individual unipotent progenitors lacking a connection on the UMAP dimension as recently suggested[37,45,46]. When the lineage gene sets were compared between the two species, both commonalities and differences were apparent. Recently, scRNAseq data of dairy cattle mammary gland was reported[50] and those from other organisms could also appear in the near future. The gene lists obtained in this study would be useful to explore core gene sets for the function of the mammary gland and the differentiation machinery, together with interspecies differences and their biological meanings.

The analysis of the TCGA breast cancer data set in light of the lineage trajectory indicated that there is no single tumor precursor for all human breast tumors as previously suggested. LumA-type tumors are thought to originate from the mature L-Hor lineage, which aligns with previous studies of FACS-enriched ML cells and the original scRNAseq of the human breast epithelium[7,37]. Our results support a possibility that LumB-type cancers are from immature hormone-sensing lineage cells. The unfavorable prognosis of LumB cancer may be associated with the increased proliferation and stemness inherent to the progenitor population. Meanwhile, the majority of basal tumors showed higher Stem and Alv scores, which suggests their origins in immature L-Alv cells. The concept linking TNBCs and the Alv lineage is not new and has been proposed in *BRCA*-mutant basal tumors[6,7,37,51]. *BRCA1* mutation carriers have an expanded L-Alv population[7,51], which would result in the transformation of this cell population later in life. Among TNBCs, the LAR subtype has been associated with androgen receptor expression and luminal

lineage gene signature[52]. Such tumors may lose ER and progesterone receptor (PR) expression during transformation from hormone-sensing cells. Contrary to *BRCA1*, research on the effects of *BRCA2* mutations are currently limited[53,54]. Future studies should aim at possible dysregulation of hormone-sensing lineage cells in *BRCA2* mutation carriers as suggested in this study.

MaSCs/basal lineage has been suggested as cells of origins for some breast cancers in mouse models[6,55]. However, little is known about the true incidence of the MaSC-driven tumors in humans, except for the finding that the transformation of CD10[+] human basal/myoepithelial cells can mimic the rare, metaplastic, and claudin-low-type cancer[6,56]. Consistent with this observation, we confirmed that some Basal and Normal-like tumors had increased MaSC/basal lineage features, and that they all had been diagnosed as metaplastic carcinomas. They also shared claudin-low tumor features. Collectively, our results underpinned the idea that MaSC/basal cells are cells of origin of metaplastic, triple-negative cancers. Lastly, to our best knowledge, almost nothing is known about the cells of origin of Her2 cancer. Our analyses reveal that HER2 tumors have features of the L-Hor cell lineage and are indistinguishable from luminal-type tumors, especially LumB tumors in terms of lineage features. This finding may explain the existence of HER2+ and HR+ double-positive LumB subtype. In other words, they may share progenitor cells in hormone-sensing lineage as their cells of origin, but the driving mutation shapes their clinical phenotypes.

The examinations of the gland reorganization and its relationship to breast carcinogenesis revealed a striking difference in the gland composition between adult/postmenopausal glands and glands at pregnancy, which corresponded to the human breast cancer epidemiology; pregnancy-associated breast cancer is characterized by a higher TNBC (35.9–48.4%) subtype rate compared to the other life stages[57–59]. This could be attributable to increased L-Alv lineage cells in the pregnant gland (Fig. 4c). During the perinatal and pubertal stages, the gland has more LPs, which are more proliferative and considered to be cells of origin for multiple types of breast cancer (Fig. 4b). The relationship between these characteristics and vulnerability to external exposures needs to be further determined on a per-subject basis[8]. Another important epidemiological observation is that parity is associated with a decreased risk of breast cancer incidence[60,61]. We observed a decrease in the luminal population and the general stemness of the gland during lactation compared to that at pregnancy. A recent study suggested that the protective effect of parity against breast cancer was brought at a specific point (34 weeks in humans) during pregnancy[62]. Future integration of scRNAseq data from multiple time points during pregnancy and lactation would help with understanding the mechanism more precisely.

Epidemiologically, the inclusion of progesterone in hormone replacement therapy (HRT) after menopause increased the breast cancer risk compared to estrogen only[17,63]. Our study supported this by finding that the addition of progesterone to estrogen further expanded the luminal epithelial cells, especially alveolar cells. Although an increase in progenitor population was not clear during sampling in our analysis, this could be due to the relatively lower doses and short half-life of progesterone in the blood[64,65]. The recent reports from the Women's Health Initiative indicated that the combination HRT increases the risk for both hormone receptor-positive and -negative breast cancers, whereas estrogen-only HRT decreases the overall breast cancer incidence[66,67]. Partial inconsistency between the epidemiology and our analysis could be due to ignorance of external hormonal effect on pre-exsisting lesions. Expansion of the L-Alv cells was also observed in glands during pregnancy and the diestrus phase, accompanied by RANKL elevation when the circulating progesterone level was known to increase[68].

PBDEs, which were once widely used as flame retardant, have been reported to activate ER in vitro and further promote estrogen-induced mammary gland regrowth in the 10-week postmenopausal model[9,18]. However, PBDE exposure seemed to partly inhibit gland regrowth by hormones in this 20-week postmenopausal model, doubling the hormone deprivation time. One possibility is that PBDEs, which are weak ligands of ER, would competitively inhibit the binding of estrogen to its receptor as the deprivation period increases[63]. It is also possible that binding of PBDEs to multiple receptors, including progesterone receptor and aryl hydrocarbon receptor, caused the complex response with both epithelial and non-epithelial components[9,18]. Although further studies are needed to define the mechanisms, our serial studies expand our knowledge on the endocrine-disrupting chemical- and HRT-induced reorganization of the mammary epithelium at a single-cell resolution.

Although our results revisited, confirmed, and are also supported by multiple previous milestone studies, the limitation of the study is its computational nature. The inferred cells of origin, reorganization during different life stages, and the associated risk increase for the specific types of breast cancer need to be determined on per-subject basis. Especially, the existence of bipotent LP cells in fetal and adult glands should be explored further. The data-filtering process removed a considerable number of cells, or even an entire sample, due to the presence of putative multiplets and low-quality cells. This should be carefully interpreted and revisited, because the analytical pipeline of scRNAseq is still in its early stage. Although the human lineage genes were validated across two different scRNAseq modalities, the data came from only seven individuals in one study. The analysis of human data, including the TCGA data set, should be discussed with caution until additional relevant human scRNAseq data becomes available and refines the lineage-specific gene sets.

In conclusion, we constructed a mammary cell gene expression atlas and defined the lineage-specific gene sets to infer the location of the given cell population on the trajectory. Our results revisited and added insights to the relationship between the cellular hierarchy in the gland and the development of the specific breast cancer subtypes. The catalog of identified gene/pathway lists and the integrated data are fully accessible in the Supplementary Data, at the UCSC Cell Browser website (https://mouse-mammary-epithelium-integrated.cells.ucsc.edu), or in a data repository[69], which could be a good resource in the mammary gland development and carcinogenesis fields.

## Methods

**Chemicals**. PBDEs (2,2',4,4'-tetrabromodiphenyl ether (BDE-47), 2,2',4,4',6-pentabromodiphemyl ether (BDE-100), and 2, 2',4,4'5,5'-hexabromodiphenyl ether (BDE-153)) were purchased from AccuStandard, Inc., New Haven, CT. These were three major PBDE congeners detected in women from the California Teachers Study[70]. 17β-Estradiol, progesterone, and dimethyl sulfoxide (DMSO) were purchased from Sigma-Aldrich Corporation, St. Louis, MO.

**Animal**. Eight-week-old female BALB/cj mice were purchased from the Jackson Laboratory (BarHarbor, ME) and housed in AAALAC (Association for Assessment and Accreditation of Laboratory Animal Care International)-accredited Animal Resources Center. Animal research procedures used in this study were approved by the Institutional Animal Care and Use Committee of City of Hope and were performed according to the institutional and NIH guidelines for animal care and use. The housing environment was prepared to prevent animals from undesired environmental exposures to chemicals and materials with potential endocrine-disrupting activity[9]. Mice were housed in polypropylene cages with Sani-Chips beddings and drinking water was filtered twice using reverse osmosis and carbon block system. Corn-cob bedding was avoided due to the potential estrogenic activity.

**Menopausal HRT and PBDE model**. Mice were ovariectomized at the ninth week after birth. Twenty weeks after ovariectomy (OVX), mice were randomized [vehicle (OVX), 17β-estradiol (OVX_E2), 17β-estradiol+PBDE (OVX_E2_PBDE), 17β-estradiol + progesterone (OVX_E2_P4), and 17β-estradiol + progesterone + PBDE (OVX_E2_P4_PBDE); $n = 8$ each)] and treated for 7 days. Special food with PBDEs mixture (BDE-47: 7.1 mg/kg, BDE-100: 0.4 mg/kg, and BDE-153: 0.9 mg/kg) was prepared (Research Diets, Inc., New Brunswick, NJ) and administered orally to mimic environmental exposure of human to PBDEs via ingestion and diet[71]. The composition has been previously determined to achieve the relevant exposure with the ratio of the three congeners found in human blood (1, 0.056, and 0.126 mg/kg/day for BDE-47, -100, and -153, respectively)[9,18]. Non-PBDE groups were fed with nutrient-matched special food with DMSO (Research Diets, Inc.). 17β-Estradiol (1 μg/animal) and progesterone (0.1 mg/animal) were administered by daily intraperitoneal injection. DMSO was used as a vehicle and injected into the control groups. A day after the last injection, mice were killed to collect mammary glands.

**Mammary gland whole-mount analysis**. Glands were fixed with 10% buffered formaldehyde and delipidated with toluene for 72 h. Then, the glands were rehydrated with gradient ethanol and stained in 0.025% Toluidine Blue. Stained glands were immersed in methanol, followed by ethanol, and a 4% ammonium molybdate solution. Afterwards, the glands were dehydrated with gradient ethanol and cleared using Histoclear (National Diagnostics, Atlanta, GA) overnight. The slides were mounted with Permount Mounting Medium (ThermoFisher Scientific, Waltham, MA). Images of the entire gland were captured using Cell³iMager Duos (SCREEN Holdings Co., Ltd, Kyoto, Japan) with 20 μm-intervals for the z-axis. Subsequently, the images were segmented for TEB-Ls and ductal structures using the machine learning implementation of the instrument (Model file CC8P06004V00, SCREEN Holdings Co., Ltd). The segmented ductal structures were skeletonized and subjected to the branching analysis using the ImageJ software[72].

**Mammary gland dissociation and scRNAseq**. The lymph node was removed from the collected fourth gland. The gland was mechanically minced with a scalpel and incubated with agitation in the digestion buffer [1.5 mg/mL DNASe I (#10104159001, Millipore Sigma, Burlington, MA), 0.4 mg/mL Collagenase IV (CLS-4, Lot: 47E17528A, Worthington Biochemical Corporation, Lakewood, NJ), 5% fetal bovine serum, 10 mM HEPES in Hank's buffered salt solution] at 37 °C for about an hour until dissociated. Then, samples were pipetted and strained through a 70 μm cell strainer. Ammonium-chloride-potassium lysis buffer was used to remove residual red blood cells and dead cells were removed using Dead Cells Removal Microbeads (Miltenyl Biotec, Bergisch Gladbach, Germany) to ensure the sample viability (> 80%) for scRNAseq.

Cells were then loaded onto the Chromium Controller (10× Genomics, Pleasanton, CA), targeting 2000–5000 cells per lane. The Chromium v2 single-cell 3′-RNAseq reagent kit (10× Genomics) was used to partition cells into gel bead-in emulsions and subsequently generate the sequencing libraries according to the manufacturer's protocol[73]. The libraries were sequenced with a Hiseq 2500 (Illumina, San Diego, CA) with a depth of 50k–100k reads per cell. Raw sequencing data were processed using the 10× Genomics Cell Ranger pipeline (version 2.0) and aligned to the mm10 mouse genome.

**Data analysis environment**. The subsequent computational analyses were performed in the following environment: R was run using RStudio Desktop in Windows 10 (ver. 1803)[74,75] and Python was run using JupyterLab in Ubuntu (ver. 18.04 LTS) built on Windows Subsystems for Linux[76,77]. The version information and availability of the softwares are summarized in Supplementary Table 1.

**Five scRNAseq data sets of the mouse mammary gland**. A database search was performed to identify scRNAseq data of mouse mammary gland generated using the droplet-based scRNAseq technique (10× chromium system) (as of September 2019)[73]. The data sequenced on other platforms were excluded to avoid potential differences inherent to the techniques that would influence the data integration process. As a result, four scRNAseq data sets were identified[10–13]. Giraddi et al.[10] collected mammary glands at different developmental stages ranging from embryos at pregnant day 16 to adult virgin mice. Pal et al.[11] studied pubertal glands and adult glands at different estrus statuses. Bach et al.[12] analyzed samples from nulliparous, pregnant, lactating, and involuted glands. Data of scRNAseq of the adult mammary gland was also retrieved from the Tabula Muris Consortium[13]. The data from our study includes cells from surgically menopaused mammary glands, with or without treatment of 17β-estradiol, progesterone, PBDEs, or their combinations. The detailed information for each study can be found in Supplementary Data 1 and 2, and the original publications[10–13]. The number of barcodes before and after the preprocessing was also summarized in Supplementary Data 2. The life stages (embryonic, neonatal, pubertal, virgin adult, pregnant, lactating, involuted, or menopausal) were manually assigned to each sample (Supplementary Data 2).

**Quality check and preprocessing of the mouse scRNAseq data**. The analyses were performed using R and the *Seurat* package, unless otherwise specified[19,74]. The purpose of preprocessing was to remove low-quality cells, epithelial doublets (or multiplets), and stromal cells based on the number of gene features and transcripts, and the percentage of mitochondrial genes. The additional examinations included cell cycle, differentially expressed genes (DEGs), *DoubletFinder*, and correlation analyses[16,78].

As the first step, low-quality barcodes with <500 gene count (nFeature_RNA) < 1000 unique molecular identifier count (nCount_RNA), or >5% proportion of mitochondrial genes (percent.mt) (>10% for the samples from Giraddi et al.[10] and this study) were filtered. The cutoff values for percent.mt were determined on a data set basis to include the majority of cells in the data set (Supplementary Fig. 2a). The relatively higher percent.mt values in the two data sets might be due to technical differences or sample quality. These differences did not seem to influence the data integration (Fig. 1c). Then, nFeature_RNA, nCount_RNA, and percent.mt values were further compared between the data sets and the samples within each data set (Supplementary Fig. 2). The five scRNAseq data sets had a comparable number of genes expressed. The data set from Tabula Muris[13] did not have any gene expressions transcribed from the mitochondrial genome (Supplementary Fig. 2a, e). This was because the reference used for annotation in the Tabula Muris project did not contain the mitochondrial genes as explained by the authors (https://github.com/czbiohub/tabula-muris/issues/221). This feature did not seem to influence the integration process either. One basal cell sample from adult mammary glands (Adult_Basal(P)) had fewer number of genes expressed in cells compared to other samples in the same data set (Pal et al.[11] and Supplementary Fig. 2c). One possibility for this would be that fully differentiated cells (basal cells) have fewer gene expressions compared to less differentiated cells (progenitors) that are supposed to be included if the gland was digested and sequenced as a whole, such as in Adult(P). A recent paper showed such a relationship between the number of expressed genes and differentiation status[28]. However, such a drastic difference was not observed for the basal cells in the Giraddi data set [Adult_Basal (G) compared to other samples in the data set (Supplementary Fig. 2b)]. Therefore, it was concluded that the quality of the Adult_Basal(P) data in the Pal data set could be compromised. This sample was excluded from further analysis.

Following the initial filtering and quality check, each sample was processed individually. The data were normalized and scaled using standard functions in *Seurat*. Then, highly variable features were identified with the vst method. The principal component analysis was performed based on the top 2000 variable features. The UMAP dimension reduction and Louvain clustering were performed[16,20]. The number of principal components used for UMAP and clustering was empirically determined between 5 and 13 through an examination of scree plots and UMAP plots in each sample. The resolution for Louvain clustering, based on a shared nearest-neighbor graph, was also determined individually between 0.1 and 0.6. These values were summarized in Supplementary Data 2. After a sensible clustering was obtained, another quality check was performed on a cluster basis to find potential contamination by dying cells (high percent.mito), low-quality data (low nFeature_RNA and nCount_RNA values), and multiplets (unreasonably high nFeature_RNA and nCount_RNA values compared to other clusters). Then, the DEG analysis, cell cycle scoring, *DoubletFinder* analysis, visualization of established marker genes, and correlation analyses of pseudo bulk RNAseq were performed to putatively identify and, accordingly, filter out multiplets and contaminating non-epithelial cell clusters. The DEG analysis and cell cycle scoring were performed using *FindAllMarkers* and *CellCycleScoring* functions in *Seurat*, respectively. *DoubletFinder* generates hypothetical doublets from a random combination of cells in the data and compares them to the other cells to point out potential multiplets[78]. The same principal component values for the dimension reduction and pN = 0.25 were input for the *doubletFinder_v3* function. pK values were determined using the *find.pK* function in each sample (Supplementary Data 2). The estimated doublet rate in the 10× Chromium system was reported to be ~10% by the manufacturer (10× Genetics)[73]. They determined the number based on experiments in which mouse and human cell lines were dissociated, mixed, and sequenced. However, the number can be much higher (~28%) when one dissociates cells from the epithelial tissue, possibly due to the strong cell–cell adhesion, as recently reported by Wang et al.[79] using human pancreas tissue. Therefore, an estimated doublet rate for *doubletFinder_v3* was set at 20%. The following representative markers were used to categorize cells: *Epcam* (epithelial cells), *Krt14* and *Acta2* (basal epithelial cells), *Krt18* (luminal epithelial cells), *Areg*, *Esr1*, and *Ly6d* (L-Hor cells), *Csn3* and *Elf5* (L-Alv cells), *Mki67* (proliferating cells), *Fosb* (stressed cells), *Col1a1* and *Vim* (fibroblasts), *Ptprc* (hematopoietic cells), *Cd62* (macrophages), *Des* (pericytes), and *Cdh5* (endothelial cells)[9–12,80].

Collectively, putative annotations of clusters were made (Supplementary Data 3). Putative multiplets clusters were excluded from the data based on the following criteria: (1) a cluster had increased values of nCount_RNA and nFeature_RNA as a consequence of capturing >2 cells; (2) the multiplets had no distinct marker expression, which exclusively separated a cluster from others; (3) a suspected multiplet cluster was pointed out by *DoubletFinder*; and (4) the multiplets had a gene expression pattern as a mixture of other existing cell types in the tissue. When a cluster met criteria 1–3, a correlation analysis was further performed to see if the cluster also met criterion 4. To perform the analysis, the normalized gene expression in each cluster was averaged using the *AverageExpression* function and a hypothetical doublet cluster was generated by averaging gene expression of the two clusters of interest. Then, the correlation between the observed and hypothetical clusters was calculated. When the suspected

cluster and a hypothetical cluster had an extremely strong correlation (coefficient ≥ 0.99), it was considered to be a multiplet cluster and was then removed from the following analysis. Contaminating non-epithelial cells were also excluded based on marker gene expressions (Supplementary Data 3). After the samples were individually processed, all the samples in each data set were merged and the merged data went through the same process to filter out rare multiplets or contaminating cells that would not stand out as a distinct cluster in each sample due to the small number of cells (Supplementary Data 3). In this second processing, the estimated doublet proportion for *DoubletFinder* was decreased to 10%, assuming that most of the doublets would have been removed in the first processing. The cleaned data by this two-step preprocessing was used for data integration.

**Data preprocessing example 1: Adult(P) sample from the Pal et al. data set**. The raw data included Lin− (Ter119−, CD31−, CD45−) CD24+ mammary epithelial cells from the mammary gland of the adult FVB/NJ mice ($n = 3302$) (Supplementary Data 1 and 2)[11]. The low-quality barcodes were filtered out based on the standard threshold. UMAP dimension reduction and Louvain clustering were performed with ten principal components and at a resolution of 0.1, respectively, which resulted in seven distinct clusters (Supplementary Fig. 3a). Among them, cluster 3 (C3) drew attention as potential multiplets due to its higher nFeature_RNA and nCount_RNA values when compared to the other clusters (Supplementary Fig. 3a). The DEG analysis revealed the distinctive gene expression profile between clusters, except for C3. The C3 cluster showed a mixed gene expression, similar to both C1 and C2 (Supplementary Fig. 3b). Although proliferating cells can have increased gene expression associated with cell division[81], the cell cycle analysis did not point out C3 as proliferating cells (Supplementary Fig. 3c). Besides, the *DoubletFinder* analysis indicated C3 as a potential doublet cluster with a pK value for the estimation of 0.5% (Supplementary Fig. 3d). By checking the well-established marker genes for the mammary epithelium, C0, C1, and C2 were putatively identified as basal epithelial (*Krt14+ Acta2+*), L-Hor (*Epcam+ Krt18+ Areg+ Esr1+*), and L-Alv (*Epcam+ Krt18+ Csn3+ Elf5+*) cells, respectively (Supplementary Fig. 3b, e)[10–13]. However, C3 showed expression of both L-Hor and alveolar markers. According to these observations, a hypothetical doublet cluster was generated by averaging the mean gene expressions of C1 and C2, and a correlation analysis was performed. As a result, the gene expression patterns of C3 and the hypothetical cluster (C1 + C2) were demonstrated to have had a strong correlation with the correlation coefficient of 0.99 (Supplementary Fig. 3f). Collectively, C3 was determined to be doublets derived from cells in C1 and C2, and therefore were excluded from further analysis.

An estimated doublet rate based on the equation[79] of doublet rate = (frequency of C3)/(2 × (frequency of C1) × (frequency of C2)) was 36.5%. This is much higher than the <10% multiple cell incorporation rate by chance in the Chromium system, but is still comparable to the doublet rates from the scRNAseq analysis of the human pancreas (~28%)[73,79]. This could be attributed to insufficient dissociation of the tissue rather than the random incorporation of multiple cells into a single Gel Bead-in-Emulsion. These specific cell types (hormone receptor-positive sensing cells and hormone receptor-negative alveolar cells) are located adjacent to each other, forming epithelial tight junctions in the mammary gland, which could be resistant against enzymatic dissociation[2,82]. Supporting this idea, there are not as many basal–luminal or epithelial–non-epithelial doublets as there are L-Hor-alveolar doublets (Supplementary Data 3).

Further, C4 was identified as pericyte/endothelial (*Des/Cdh5+*), and C5 and C6 were considered to be fibroblast (*Col1a1+ Vim+*) clusters (Supplementary Fig. 3g and Supplementary Data 3). In conclusion, C3–C6 were removed from the analysis and C0–C2 were retained for the second preprocessing.

**Data preprocessing example 2: OVX_E2(S) sample from our study**. The sample was generated from the adult mammary gland of the ovariectomized BALB/cj mice treated with 17β-estradiol for 1 week. The raw data included 2229 barcodes. After the removal of low-quality barcodes, cells were clustered and visualized on UMAP dimensions with the top ten principal components and a resolution of 0.5. Ten distinct clusters were identified, and C2 and C6 had higher nFeature_RNA and nCount_RNA values compared to the others (Supplementary Fig. 4A). As the scRNAseq was performed without cell sorting, more non-epithelial clusters were present in the sample (Supplementary Fig. 4b).

Based on the gene expressions, cell cycle analysis, *DoubletFinder* (pK = 19%), and the correlation analysis, C2 was considered to be doublets from L-Hor (C1) and L-Alv (C5) cells (Supplementary Fig. 4b–f). Although C6 had a similar gene expression to C2, C6 consisted of proliferating cells (G2M/S) and had unique marker gene expressions, such as *Mki67* (Supplementary Fig. 4g). Therefore, C6 was retained and labeled as proliferating luminal cells. Other clusters were labeled as basal cells (C7; *Krt14+*), fibroblasts (C0, C3, and C4; *Col1a1+ Vim+*), hematopoietic cells (C8; *Ptprc+ Cd52+*), and endothelial cells (C9; *Cdh5+*). C1, C5, C6, and C7 were retained for further analysis.

**Mouse scRNAseq data integration using Seurat v3 algorithm**. After preprocessing and before integration, the five data sets were merged, clustered, and visualized on UMAP dimensions (Supplementary Fig. 5a). Pseudo-RNAseq was

also performed by averaging the gene expression of each sample, which was then followed by a principal component analysis. The clusters were mainly separated by the data sets on both UMAP and principal component plots rather than the sample features (Supplementary Fig. 5a, b). These "batch effects" were irrespective of the basic quality parameters of the data sets, such as nFeature_RNA, nCount_RNA, or percent.mt (Supplementary Fig. 5a, b), and therefore considered to be inherent to the techniques or conditions in each study. This confirmed an essential need for an adequate data integration algorithm before any comparisons and analyses beyond each data set could be made.

The integration of data was performed using *Seurat v3* according to the developers' vignette[19]. Briefly, each data set was individually log-normalized and the highly variable features were identified. Then, the 2000 integration anchors were identified using the *FindIntegrationAnchors* function with the dimensionality of 50 based on the default setting and the developers' recommendations. Data integration was performed with the *IntegrateData* function using a different data set, combinations of multiple data sets, or none as the reference data set(s). The resulting integrated data were visualized on the UMAP spaces using the top ten principal components (Supplementary Fig. 5c), which revealed the consistency of the structures of the integrated data; three major tip clusters were connected at the center and the very central part of the structure was composed of embryonic cells from Giraddi et al.[10] in most cases. The representative integrated data were chosen[11] for further analysis in this study. The integrated data were scaled, clustered, and visualized with the routine processing (Fig. 1 and Supplementary Fig. 6). Pseudo-RNAseq was also performed using the integrated data, to confirm an effective removal of the batch effect (Supplementary Fig. 5b). For three-dimensional plotting, UMAP dimension reduction was performed in a three-dimensional space and the result was plotted by *plot_ly* package (Fig. 1b).

**CytoTRACE analysis**. The *CytoTRACE R* package was used for this purpose[28]. Three thousand cells from each cluster was randomly chosen in the mouse integrated data. The original five data were subset for the selected cells, merged with Scanorama-based integration, and evaluated regarding the GCS and CytoTRACE score (iCytoTRACE). For the human normal breast data, 15,000 cells were randomly selected from the integrated data and evaluated with the iCytoTRACE function. Besides, the entire data from Giraddi et al.[10] and Bach et al.[12] were individually scored using the *CytoTRACE* function, to confirm the results in the non-integrated data. The barcode samplings were performed due to the limitation of the computer memory. The results were visualized on UMAP and by box plots grouped by the different life stages (Supplementary Fig. 7a, b). In this case, the sample that only included the specific cell types [Adult_Basal(G)] or the samples that received the HRT treatment [OVX_E2(S), OVX_E2_PBDE(S), OVX_E2_P4 (S), and OVX_E2_P4_PBDE(S)] were excluded.

**Data integration by Harmony, LIGER, and scAlign**. The integration was performed according to developers' vignettes (*Harmony*[29], https://github.com/immunogenomics/harmony; *LIGER*[30], https://github.com/welch-lab/liger; *scAlign*[31], https://github.com/quon-titative-biology/scAlign). For integration using *Harmony*, the cleaned data sets were merged, normalized, and scaled, and the top 50 principal components were calculated using 2000 variable features with *Seurat*. The cells were then integrated and plotted on harmony dimensions based on the principle components using the *RunHarmony* function with the default settings of the *harmony* R package[29]. The data were visualized on a UMAP plot using the top six harmony dimensions. For data integration using *LIGER*, the cleaned data sets were merged, normalized, and then individually scaled using *Seurat*. Joint matrix factorization was performed using *RunOptimizeALS* in the *liger* R package[30] with the default values of $k = 20$ and $\lambda = 5$. Subsequently, Quantile normalization and joint clustering was calculated using the *RunQuantileNorm* function with its default settings. The results were summarized on a UMAP plot using the top 20 iNMF dimensions. In *scAlign* integration, the cleaned data sets were individually normalized and scaled, and variable genes were calculated using *Seurat*. Then, common variable genes across the five data sets were identified and a canonical correlation analysis was run by *scAlignCreateObject* in the *scAlign* R package[31]. Subsequently, the data were aligned by *scAlignMulti* with the default settings. UMAP was calculated using the top 64 aligned multi-Canonical Correlation Analysis (CCA) dimensions.

**The STREAM analysis of the mouse mammary epithelium**. The STREAM algorithm is a trajectory inference method that can robustly reconstruct developmental trajectories with accurate pseudotime estimation from scRNAseq data[33]. As the mapping function of STREAM cannot introduce new fate branches in addition to the ones present in the reference principal tree, an initial trajectory is needed to include all the states in the mammary gland[33]. We chose the data from Giraddi et al.[10] for the initial trajectory reconstruction. We chose the data from Giraddi et al.[10] for the initial trajectory reconstruction, as the data included all the cell fates identified in the integrated analysis from the embryonic true MaSCs to the three differentiated cell types. Using the *stream python* package, the cells were projected to the Modified Locally Linear Embedding spaces considering the 2000 genes used for the data integration by Seurat. Then, the Elastic Principal Graph trajectory was seeded, adjusted, and optimized using *seed_elastic_principal_graph*, *elastic_principal_graph*, and *extend_elastic_principal_graph* functions, respectively, with

recommended parameters (Supplementary Fig. 8a). Visualization on stream plots was performed with the *stream_plot* function (Supplementary Fig. 8a). The learned trajectory was used as the initial trajectory.

Next, all the cells in the integrated analysis were projected on the initial trajectory using the same 2000 anchor genes for the integration with the *map_new_data* function to map the cells on the one consensus trajectory and accordingly calculate pseudotime values (Supplementary Fig. 8b). The results were summarized according to the data sets, stages, mouse strains, and samples (Supplementary Fig. 8c, d). For visualization in the main figure, 500 cells were sampled from each cluster and 2 embryonic samples (Fig. 2a, b). DEG analysis and transitional gene analysis for each leaf branch (S0_S1, S0_S2, S3_S4, and S5_S3) were performed using this sampled data with the *detect_leaf_genes* and *detect_transition_markers* functions, respectively. The top transition genes were visualized with *pheatmap* and subjected to enrichment analysis for the Hallmark, C2, and C5 gene sets available at the MSigDB using the *enricher* function in the *clusterProfiler* R package (Supplementary Fig. 8e and Supplementary Data 5)[83]. For the visualization of the selected samples on the trajectory, a backbone trajectory was generated by projecting 100 cells from each cluster in the integrated data (Fig. 4b, c).

**Identification of lineage-specific genes**. The STREAM analysis output cell–cell distance on the trajectory for any given pairs. This enabled us to obtain the pseudotime of any given cell starting from any given leaf state. Accordingly, we calculated pseudotime from S5, S4, S2, and S1 for all the cells and defined them as "Stem," "Basal," "Alv" (L-Alv), and "Hor" (L-Hor) pseudotime, respectively (Fig. 2c). Then, we calculated correlations of gene expressions with each pseudotime. The five studies were analyzed individually. Spearman's rank correlation coefficient was calculated between the scaled gene expression and the pseudotime, and the resulting correlation coefficients for a given pair of gene and pseudotime were averaged from the five studies. The results are shown in Supplementary Fig. 9a, b and summarized in Supplementary Data 6. For example, *Mif* and *Krt18* had the strongest positive and negative correlation with S5 pseudotime ("Stem"), respectively (Supplementary Fig. 9a). Although the behaviors of the single genes could predict differentiation status to some extent, we tried to improve the prediction performance by generating lineage-specific gene sets with the top-correlated genes, because gene set-based scoring is, in general, more robust to background noises and have been shown to outperform single gene expression in terms of cell-type identification in scRNAseq data[84].

**The lineage-specific gene sets and their performance evaluation**. An empty gene set was generated for each pseudotime and the top-ranked genes were incorporated into the gene set in the order of their correlation ranking in increments of one gene up to 100, then 20 up to 1000. After each incorporation, the performance of the gene set was evaluated by scGSVA[34] (*GSVA* R package) and Spearman's rank correlation analysis with pseudotime. We expected that the performance of the gene set would initially increase as the number of genes in the gene set increased and then reach a plateau or start to decrease at some point as the genes with lower correlation to the pseudotime started to be incorporated. In favor of that expectation, the performance of the gene sets plateaued or started to decrease for all of the four pseudotimes (Supplementary Fig. 9c). Similar trends were observed in each of the five data sets investigated. The best performing gene sets included the top 160, 240, 500, and 200 genes for "Stem" (S5 pseudotime), "Basal" (S4 pseudotime), "Alv" (S2 pseudotime), and "Hor" (S1 pseudotime) states, respectively (Fig. 2c and Supplementary Fig. 9c). The correlation of the curated data sets with the corresponding pseudotime outperformed that of single gene expression in terms of the correlation coefficient (Supplementary Data 6 and 7).

Then, we compared the performance of the curated gene sets with other existing RNA-based features. A thousand cells were sampled from each study and scGSVA scores for the gene sets available at the MSigDB ($n = 22,540$; as of 3-20-2020, https://www.gsea-msigdb.org/gsea/index.jsp) were calculated. For the retrieval of the data adjusted for the mouse transcriptome, the *msigdbr* R package was used. Afterward, Spearman's rank correlation coefficients of the pseudotime were calculated to the GSVA scores, CytoTRACE, GCS, and the other basic characteristics of each barcode (nFeature_RNA, nCount_RNA, and percent.mt). The results were averaged between the studies and ranked accordingly (Fig. 2c, Supplementary Data 7, and Supplementary Fig. 10a–c).

**The lineage-specific gene set-based data summarization**. The scGSVA scores for the four curated gene sets were calculated for the five data sets using individually scaled data. Then, the results were combined and projected to lower dimensions using the *UMAP* R package. To reduce the computational burden for the summarization of the 50 K cells, UMAP dimension reduction was performed using the cells from Giraddi et al.[10] and then all the cells were projected on the constructed UMAP dimensionality with the *predict* function (Fig. 2c and Supplementary Fig. 11a). For the visualization on the ternary plots, the *ggtern* R package was used (Supplementary Fig. 11b).

**Analysis of human breast epithelium scRNAseq data**. scRNAseq data of the breast epithelium from four healthy adults were retrieved from a public database[37]

(Supplementary Data 2). The data from each sample were individually processed and annotated according to the original paper (Fig. 3a, Supplementary Fig. 12a, b, and Supplementary Data 3). The original authors identified five distinct clusters, which was simplified in this analysis to major three clusters by merging B and Myo clusters into B/Myo, and L1.1 and L1.2 clusters into L1.1/L1.2. The following representative markers were used to identify cells: *KRT14* and *ACTA2* (basal epithelial cells), *KRT18* (luminal epithelial cells), *AREG* and *ANKRD30A* (L2 cells or L-Hor cells), *SLPI* and *ELF5* (L1.1/L1.2 cells or L-Alv cells), *MKI67* (proliferating cells), and *COL4A1* and *VIM* (fibroblasts). The human gene symbols were converted into the corresponding mouse gene symbols using the Ensembl database (Release 99; https://uswest.ensembl.org/index.html)[85] for the label transfer analysis (Fig. 3b and Supplementary Fig. 12c). Then, CCA-based transfer anchor identification was performed with the annotated mouse integrated data set as a reference data (*FindTransferAnchors*). Identities of human data were predicted using the 2000 transfer anchors and the dimensionality of 50 (*TransferData*).

Data integration were using *Seurat_v3* algorithm, as performed in the mouse data integration (Supplementary Fig. 13a, b). No reference data were set. The integrated data were projected to a lower UMAP dimension using the top ten principal components. For the CytoTRACE analysis, 15,000 cells were sampled from the total barcode pool and the *iCytoTRACE* analysis was run using the individual.

For the STREAM analysis, 1000 cells were sampled from each individual and the base trajectory was built using the top 10 principal components from the integrated data (Supplementary Fig. 13c, d). The top PCs were selected as an input, instead of integration anchors as in the mouse data analysis, because we did not intend to visualize the expression of each gene on the trajectory and therefore tried to decrease the computational burden. Then, all cells were mapped onto the base trajectory and pseudotime was calculated for each leaf node.

The top-correlated genes and the lineage-specific gene sets were identified and evaluated for the S4 ("Basal"), S2 ("Alv"), and S1 ("Hor") pseudotimes (Supplementary Data 8 and 9, and Supplementary Fig. 13e, f). The curated gene sets outperformed the other RNA-based features, except for the one that slightly outperformed the curated "Basal" gene set. The gene set was identified as commonly upregulated genes in FACS-sorted MaSC/basal cell types from both mouse and human mammary tissues[36]. The other gene sets identified in the same study also appeared among the top gene sets for the S2 and S1 pseudotimes, which strengthened the accuracy of our annotation. For consistency, the curated "Basal" gene set was used for the subsequent analysis.

Using the curated gene sets, the scaled data were scored with scGSVA analysis. Then, the results from the 24 K barcodes were summarized using a UMAP dimension reduction (Fig. 3c and Supplementary Fig. 14a–c). The performance of "Stem" gene sets adopted from the analysis of the mouse data was also evaluated in comparison with the unbiased stemness inference by the CytoTRACE algorithm (Supplementary Fig. 14d). The scores for the three differentiation gene sets were also visualized on ternary plots (Supplementary Fig. 15a).

For further evaluation of the obtained gene sets, the additional data from the three individuals sequenced using Fluidigm C1 were retrieved[37]. The data were integrated and definitively annotated according to the original publication (Supplementary Fig. 15b). Then, the scaled gene expression data were scored using the human lineage-specific gene sets and visualized on ternary plots (Supplementary Fig. 15c).

To gain insight about interspecies commonalities and differences, the human and mouse lineage-specific gene sets were compared, and the common and species-specific genes were identified (Supplementary Fig. 15d and Supplementary Data 10). They were also subjected to enrichment analysis for the Hallmark, C2, and C5 gene sets available at the MSigDB (Supplementary Data 11).

**Adaptation of the lineage-specific gene set-based lineage inference to bulk RNAseq data**. Aiming to validate the feasibility of the lineage-specific gene sets in the bulk RNAseq data, serial simulation analyses were performed. First, varying numbers of cells (10–1000) were sampled from each cluster in the integrated mouse data and normalized gene expression of the sampled barcodes was averaged using the *AverageExpression* function in the *Seurat* R package. To mimic the actual mammary epithelia, the sampling was also performed from the mixed barcode pool (500 barcodes were pooled from each cluster) ("Mixture"). For each condition, sampling was performed five times. Then, the data were scored using the scGSVA analysis for the three differentiation gene sets ("Basal," "Alv," and "Hor") and plotted on a ternary plot (Supplementary Fig. 16a).

Next, we evaluated the effect of stromal cell contamination. For this purpose, the Tabula Muris data set[13] was used, as the data were generated without any sorting processes and hence included stromal cells. Although our data also included a stromal population, our mice were ovariectomized and may not accurately reflect the representative population in the gland. The two samples from the adult virgin mammary glands were merged, projected to a lower dimension space, and clustered using the top ten principal components and a resolution of 0.1 (Supplementary Fig. 16b). Considering the annotation in the original publication and our integrated analysis (Supplementary Fig. 16b), C3, C7, and C8 were defined as Basal, Alv, and Hor clusters, respectively, in the following analysis. One hundred cells were sampled from C3 ("Basal"), C7 ("Alv"), C8 ("Hor"), C3 + C7 + C8 ("Mixture"), and the rest of the barcodes from the data ("Stroma"). Then, "Stroma"

cells were combined with "Basal," "Alv," "Hor," or "Mixture" cells with the varying percentage of epithelial cells in the data ranging from 50% to 100%. The sampling was performed ten times for each condition and the data of the pooled cells were averaged to make pseudo bulk RNAseq data. Subsequently, the data were scored using the scGSVA analysis for the four gene sets. The results were visualized on a ternary plot using the scores for the three differentiation gene sets (Supplementary Fig. 16c). The changes in the scores with increasing stromal contamination were also individually examined (Supplementary Fig. 16d).

Lastly, the mouse and human FACS-sorted microarray data of the mammary epithelium were retrieved to access the applicability of the method in the bulk transcriptome analysis. In the original study, the three distinct populations (MaSC, LP, and ML) were isolated from the mouse and human mammary epithelium using a panel of antibodies (EpCAM/Cd49f and CD24/CD29/CD61, respectively) followed by microarray analysis[7,36]. MaSC, LP, and ML populations are considered to correspond to the Basal, Alv, and Hor lineages, respectively, in light of the recent mammary gland biology[2,37]. The mouse and human data were scored with scGSVA analysis using the three differentiation gene sets ("Basal," "Alv," and "Hor") generated from the mouse and human scRNAseq data.

**Analysis of the TCGA BRCA data**. The RNAseq data and the clinical information of the breast cancer in TCGA (https://www.cancer.gov/tcga) was retrieved through the National Cancer Institute Genomic Data Commons (NCI GDC v22.0; accessed on 3-20-2020)[39] using the *TCGAbiolinks R* package[86]. The fragments per kilobase of transcript per million mapped reads upper quartile (FPKM-UQ) values were used for the principal component analysis and UMAP dimensionality as well as calculation of the scGSVA scores for the curated gene sets. The molecular sub-typing, histological classification, age at diagnosis, and the stromal proportion in the tissue were also retrieved from the NCI GDC database. The results of TNBC subtyping were retrieved from a previous publication[87]. The *BRCA* status was also retrieved from a literature[88]. The UMAP dimension reduction was performed using the top ten principal components, which considered all the transcripts in the data (Supplementary Fig. 17a). The data were scored with scGSVA analysis using the human lineage gene sets (Supplementary Figs. 17 and 18). The data were also scored using the DEGs from the mouse integrated data (Supplementary Data 4) for the progenitor gene signatures (Supplementary Fig. 17d).

**Analysis of the human breast cancer scRNAseq data**. The expression data and metadata were retrieved according to the original publication[43]. The data included the scRNAseq results of breast cancer from the 11 patients generated with the Fluidigm C1 system[43]. The data processing was performed according to the original metadata and non-tumor cells were excluded from the analysis, which resulted in the complete removal of one patient's (BC09) data. The data were visualized in the UMAP dimensionality using the top five principal components from the whole transcriptome (Supplementary Fig. 19a). The PAM50 molecular subtyping, scGSVA analysis, and data summarization using the scGSVA scores were performed (Supplementary Fig. 19b, c).

**Molecular subtyping**. PAM50 and claudin-low molecular subtyping was performed using *genefu R* package according to the developers' vignette[89]. For normal mammary epithelium scRNAseq data, the normalized data was used for the PAM50 subtyping on a single-cell basis. For the breast cancer scRNAseq, the averaged expression of the normalized data of the tumor cells from the individual patient was used for the PAM50 subtyping. For the TCGA data, the PAM50 classification was retrieved from NCI GDC and claudin-low subtyping was performed using the FPKM-UQ values.

**UCSC cell browser**. The data for submission at the UCSC Cell Browser were prepared according the developers' vignette (https://cellbrowser.readthedocs.io/installation.html)[32]. Briefly, the data were exported by the *ExportToCellbrowser* function in the *Seurat* R package. For the *Seurat v3* and *scAlign* data, transformed gene expression matrices (2000 and 411 genes) after integration were submitted. For the *Harmony* and *LIGER* data, scaled gene matrices in *Seurat*, including all the genes detected, were submitted. The data were available and interactively explorable at https://mouse-mammary-epithelium-integrated.cells.ucsc.edu. The *Seurat* objects for the data are also available on the website.

**Statistics and reproducibility**. All the statistical analyses were performed in R. Wilcoxon rank-sum test was used to compare the distribution of two samples. Cliff's delta values were calculated to estimate the effect size. Spearman's rank correlation coefficient was calculated for correlation analysis. $p$-Value < 0.05 was considered as statistically significant. The error bars represent SDs. The number of samples for statistical tests can be found in the figure legends where applicable. The sample size of the study was determined based on the previous study[9–13]. To verify reproducibility of the key data integration process, we input various parameters for the process and used the four different algorithms. Regarding our in vivo experiments, the model was verified by the previous publication[9] and results were supported by the data from others through the integration analysis, although the replication of the entire experiment including scRNAseq was not feasible due to its experimental burden.

**Reporting summary**. Further information on this research is available in the Nature Research Reporting Summary linked to this article.

## Data availability
The authors declare that all data supporting the findings of this study are available within the article, the Supplementary Data, and the data repository or from the corresponding author upon reasonable request. The data from the Tabula Muris Consortium was available in the Figsharewith the identifier doi.org/10.1038/s41586-018-0590-4[13,90]. The other publicly available scRNA data sets were retrieved from the Gene Expression Omnibus under the following accession codes: GSE111113 (Girradi et al.[10]), GSE103275 (Pal et al.[11]), GSE106273 (Bach et al.[12]), GSE113197 (Human normal breast, Nguyen et al.[37]), and GSE75688 (human breast cancer, Chung et al.[43]). The scRNAseq data obtained in this study were deposited in the Gene Expression Omnibus along with their associated metadata (GSE149949). The integrated data are explorable on the web browser and can be downloaded as *Seurat* R objects at https://mouse-mammary-epithelium-integrated.cells.ucsc.edu. The Mouse and human FACS-sorted microarray data of the mammary epithelium were also retrieved from the GSE under the code GSE19446 and GSE16997, respectively[7,36]. The TCGA breast cancer data were retrieved from the NCI GDC (https://www.cancer.gov/tcga)[39]. The data and custom codes in this study were deposited and available in Zenodo (https://doi.org/10.5281/zenodo.4674274)[69].

## Code availability
The custom computer scripts and the relevant data are available in Zenodo (https://doi.org/10.5281/zenodo.4674274)[69].

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

## Acknowledgements

As part of the Breast Cancer and Environmental Research Program, this work was supported by National Institute of Environmental Health Sciences U01ES026137-01 (MPIs S.C. and S.L.N.) and a pilot grant from the City of Hope Center for Cancer and Aging. We thank the City of Hope Integrative Genomics Core, which is supported by the National Cancer Institute of the National Institutes of Health under award number P30CA033572, for the excellent technical support.

## Author contributions

K.S., N.K., S.L.N., and S.C. designed research. G.C., N.K., L.B., and D.H. performed animal experiments. X.W. and J.W. performed scRNAse. K.S., G.C., and X.W. performed bioinformatics analyses. S.C. supervised research. K.S., G.C., D.H., and S.C. wrote the paper manuscript and all authors discussed the results and provided comments and feedback.

## Competing interests

The authors declare no competing interests.
