## [Transparent Peer Review File · Communications Biology]

Reviewers' comments:

Reviewer #1 (Remarks to the Author):

In this manuscript, the authors describe an integrative analysis of single-cell transcriptomes during mouse mammary epithelial development from different studies, strains, and across multiple stages. Based on their results, they define gene sets corresponding to putative mammary stem cells (MaSCs), basal cells, and two luminal populations. These gene sets are used to explore scRNA-seq data of human mammary epithelium and bulk RNA-seq profiles of breast tumors generated by The Cancer Genome Atlas (TCGA). The authors also explore the effect of development, pregnancy, and hormone replacement therapy on the populations in the gland and their implication for the risk of specific breast cancer subtypes.

Overall, the data processing steps are reasonable, and the authors have used accepted and standard approaches for filtering, visualization, and trajectory inference. The visualizations are appealing and corroborate the presented conclusions. The genes selected are also appropriate. While many of the specific findings are largely confirmatory, this work nicely synthesizes existing and newly generated data into the first global single-cell atlas of the life history of the mouse mammary gland, including branches and cell population distributions across different time points and treatment conditions. In addition, a key finding appears to be that the bipotent progenitors present in embryogenesis are not present in postnatal mammary tissue, providing additional evidence that downstream lineages are likely maintained by unipotent progenitors. These cells also appear to be lacking from mature human mammary epithelium (at least at the resolution of sequenced cells).

Despite the strengths of this work, the current manuscript suffers from several issues that dampen enthusiasm, as detailed below.

Major comments:

1. Girardi et al. (Cell Reports 2018) described a mouse scRNA-seq atlas capturing developmental stages ranging from fetal MaSCs to adult basal cells and luminal cell subpopulations. Their single-cell multilineage trajectory (Fig. 2) appears nearly identical to the ones reported by the authors. A more thorough exposition of the advantages of the data integration effort undertaken in this study would be beneficial. For example, could similar insights have been derived directly from the Girardi et al. data, or does the integrative analysis provide more robust gene signatures for the key populations? Perhaps, the authors' integrative analysis can provide new insights into the putative unipotent progenitors of each adult epithelial population? If so, this would be a welcome contribution to the field.
2. The authors used data integration to show a mapping between mouse and human epithelial populations. However, differences were also apparent. The authors missed an opportunity to delineate commonalities and differences, both for genes and pathways, which would benefit the field.
3. The authors' approach to project single-cells using gene set enrichments is not novel and has been used previously to show differentiation states in brain cancers (e.g., Fig. 2d of PMID 27806376). The authors should temper their related claims.
4. Many groups have studied the expression of mammary stem cell and epithelial population-specific genes in bulk breast tumor expression data. If candidate unipotent progenitors play a larger role than previously appreciated (e.g., in LumB or Her2 subsets, Fig. 4a,b), it would be useful to explicitly define the gene sets for candidate unipotent progenitors of each epithelial population and reassess their associations with breast cancer molecular subtypes.
5. Breast tumors have been subdivided into clinically-distinct groups that extend beyond pam50 subtypes. For example, TNBC can be subdivided into Vanderbilt subtypes. Additionally, BRCA status is a key risk factor for TNBC. How do these and other key molecular classifications of breast cancer relate to reference maps defined by the authors? Do the authors observe an association between putative cell-of-origin and age in breast cancer patients?

Minor comments:

1. Cluster 1 appears to conflate candidate unipotent progenitors of basal epithelial cells with fetal MaSCs. After refining the clustering to separate these two subsets, it will be important to determine whether any cells in the fMaSC cluster are derived from datasets other than Girardi et al. If so, this may reveal evidence for MaSCs later in development. Tracing such cells to their respective datasets and developmental time points would be warranted.
2. The authors should do a better job in the Results explaining which findings are new and which are confirmatory. The line appears to be blurred in several places.

Reviewer #2 (Remarks to the Author):

In this study the authors report the integration and reanalysis of several (embryonic and adult) mammary epithelial cell (MEC) scRNAseq studies (4 mouse and 1 human). In addition, they also include their own scRNAseq data which investigates the impact of Ovx and hormone treatment on the MEC composition. The authors then use their new integrated data to investigate the cell of origin of breast tumours using publicly available RNAseq data. Overall the authors have done a good job in combining these publicly available datasets which I believe should be made public. However, there are several parts of the manuscript where (in my opinion) the authors are over-interpreting the results. If the authors could address these issues I would support publication.

Major issues:

- 1) There are other data integration algorithms (and subsequent interpretation) other than Seurat v3. It would be good to see how the data looks like when other methods are applied. I suggest a comparison in a supplementary figure should be included at least for the overall shape of the data (eg. UMAP).
- 2) The quality control carried out by the authors excluded low quality cells from all the studies. However, according to this analysis some studies had predominantly low-quality cells (eg. Pal B. et al). This needs to be clearly highlighted in the text and implications of this needs to be mentioned with regard to the data interpretation.
- 3) The 4 mouse datasets were from different genetic backgrounds. This is a key difference and one that needs to be presented clearly. I suggest that the proportions contribution of these genetic backgrounds need to be presented for each cell population/cluster.
- 4) Line 160-164. This is a big statement regarding the bi-potent luminal progenitors. I think more analysis is needed here. The authors should show the break-down of the data from the 4 different studies and also a breakdown by mouse background.
- 5) There is a major problem with the comparison to the human dataset. First of all, there is only one study out there with scRNAseq data (there are many more in progress with much larger sample sizes) so it is premature to make any inferences based on this single study. I think the authors should remove the human analysis part and focus the manuscript on the mouse analysis.
- 6) Similarly, the comparison of human scRNAseq to the publicly available cancer datasets is premature in my opinion as the authors are relying on effectively 4 individuals from one study. They should wait till there is more data available.
- 7) It is not clear to me how the Ovx scRNAseq data fits in this paper. This feels like an add on and is also presented in a disjointed way after the human data. I would suggest that the authors either take it out or integrate it into the mouse analysis and discuss it fairly.
- 8) The discussion section needs to be toned down significantly. The authors are over interpreting their analysis. This will be made easier if the human data is taken out.
- 9) Line 408-411. Again the authors are making very serious claims by re-interpreting published data. They don't provide any new experimental evidence to support this claim. In addition, the authors fail to mention the scATACseq papers (Wahl lab) that present data which supports the presence of the bi-potent progenitor cells. A more balanced presentation of the data out there (especially if it's not their own) is required.
- 10) The combined dataset of the various mouse studies could be a nice resource for the mammary

gland community and the authors should consider generating a user friendly website to mine the data.

Responses to the reviewer 1

Remarks to the author:

In this manuscript, the authors describe an integrative analysis of single-cell transcriptomes during mouse mammary epithelial development from different studies, strains, and across multiple stages. Based on their results, they define gene sets corresponding to putative mammary stem cells (MaSCs), basal cells, and two luminal populations. These gene sets are used to explore scRNA-seq data of human mammary epithelium and bulk RNA-seq profiles of breast tumors generated by The Cancer Genome Atlas (TCGA). The authors also explore the effect of development, pregnancy, and hormone replacement therapy on the populations in the gland and their implication for the risk of specific breast cancer subtypes.

Overall, the data processing steps are reasonable, and the authors have used accepted and standard approaches for filtering, visualization, and trajectory inference. The visualizations are appealing and corroborate the presented conclusions. The genes selected are also appropriate. While many of the specific findings are largely confirmatory, this work nicely synthesizes existing and newly generated data into the first global single-cell atlas of the life history of the mouse mammary gland, including branches and cell population distributions across different time points and treatment conditions. In addition, a key finding appears to be that the bipotent progenitors present in embryogenesis are not present in postnatal

mammary tissue, providing additional evidence that downstream lineages are likely maintained by unipotent progenitors. These cells also appear to be lacking from mature human mammary epithelium (at least at the resolution of sequenced cells).

Despite the strengths of this work, the current manuscript suffers from several issues that dampen enthusiasm, as detailed below.

Authors' response:

We appreciate this reviewer's careful evaluation and valuable comments/suggestions. According to this and the other reviewers' comments, we have made significant revisions to our manuscript, including different data integration methods, additional analyses on the progenitor populations, comparisons between data from mouse and human studies, and a data deposit at the UCSC Cell Browser to allow readers to explore genes of interest interactively. We also removed overinterpretations and overstated sentences to balance our findings and those from others. We hope that our responses properly address the reviewers' comments.

Point-by-point response

#	Reviewer's comment	Authors' response
Major comment		
#1	Giraddi et al. (Cell Reports 2018) described a mouse scRNA-seq atlas capturing developmental stages ranging from fetal MaSCs to adult basal cells and luminal cell subpopulations. Their single-cell multilineage trajectory (Fig. 2) appears nearly identical to the ones reported by the authors. A more thorough exposition of the advantages of the data integration effort undertaken in this study would be beneficial. For example, could similar insights have been derived directly from	Relevant new data: Supplementary Fig. 8E and Supplementary Table 5 Thank you for this valuable comment. We recognize that the previous study by Giraddi et al. captured a very similar trajectory to the one obtained in this study. However, as indicated by this reviewer, the strength of our analysis has been that we can identify putative unipotent progenitor populations by combining the data from five studies examined at different life stages. In this revision, we explored the transitions of gene expressions through differentiation in each lineage, taking advantage of their pseudotemporal ordering. We also analyzed the transcriptomic signatures in unipotent progenitor

	the Girardi et al. data, or does the integrative analysis provide more robust gene signatures for the key populations? Perhaps, the authors' integrative analysis can provide new insights into the putative unipotent progenitors of each adult epithelial population? If so, this would be a welcome contribution to the field.	populations. As a result, we have found that the unipotent progenitors have higher expression of cell cycle and myc-related genes, which could be further explored in the future. We added the gene and pathway lists in Supplementary Fig. 8E and Supplementary Table 5, which could be useful resources to explore repopulation and differentiation machinery in each lineage. We also added the following sentences to our manuscript: (Results, Line 174) First, we focused on the putative unipotent progenitor populations (C1, C3, and C5 clusters in Fig. 1) that were not clearly identified in the previous scRNAseq studies¹⁰⁻¹³. For that purpose, we analyzed transcriptomic changes during the differentiation
--	---	---

		process in each lineage (S0_S1: L-Hor differentiation, S0_S2: L-Alv differentiation, and S3_S4: Basal differentiation). As a result, the progenitor populations were found to express genes associated with cell cycle progression and myc pathways compared to their mature counterparts (Supplementary Fig. 8E and Supplementary Table 5). (Discussion, Line 448) As a result of the filtering and integration of the multiple datasets, the trajectory obtained in the current study could separate the putative unipotent progenitor populations, which have not yet been identified in prior scRNAseq studies⁹⁻¹³. The indicated characteristics of the progenitor cells were high proliferation capacity and activation of the myc pathway, which have been repeatedly associated with progenitor populations in other tissues^{51,52}. Although
--	--	--

		further validation studies are warranted, the obtained gene and pathway lists could be useful resources to explore the repopulation and differentiation machinery in each lineage of the mouse mammary gland.
#2	The authors used data integration to show a mapping between mouse and human epithelial populations. However, differences were also apparent. The authors missed an opportunity to delineate commonalities and differences, both for genes and pathways, which would benefit the field.	Relevant new data: Supplementary Fig. 15D and Supplementary Tables 10-11 Thank you for this important comment. We compared the data between the species and found that human and mouse have different lineage gene sets despite sharing several well-known marker genes and gene signatures. We believe that these results can be good resources to study inter-species commonalities and differences. Accordingly, we added Supplementary Fig. 15D and Supplementary Table 9. We revised the text as follows:

		(Results, Line 275) When mouse and human lineage-specific gene sets were compared, both commonalities and differences were recognized (Supplementary Fig. 15D). While the known lineage markers and the relevant gene signatures were preserved in the two species, a significant number of lineage genes were species-specific (Supplementary Tables 10 and 11). (Discussion, Line 467) On the other hand, when lineage gene sets were compared between the two species, a significant number of genes were found to be species-specific. Recently, scRNAseq data of dairy cattle mammary gland was reported⁵⁵, and those from other organisms could also appear in the near future. The gene lists obtained in this study would be a
--	--	---

		foundation to explore core gene sets for the function of the mammary gland and the differentiation machinery, together with inter-species differences and their biological meanings.
#3	The authors' approach to project single-cells using gene set enrichments is not novel and has been used previously to show differentiation states in brain cancers (e.g., Fig. 2d of PMID 27806376). The authors should temper their related claims.	We appreciate this information and noticed that some of our sentences were misleading. We made changes to the text as follows: (Results, Line 204) A similar approach has been reported previously to estimate the differentiation status of human oligodendrogloma cells³⁶. (Results, Line 282) Therefore, lineage inference based on the gene sets defined in this study would predict their cells of origin as previously attempted^{6,38}.

		(Discussion, Line 597) In conclusion, we constructed a putative lineage trajectory of the mammary epithelium throughout important WOS by integration of multiple datasets and defined the lineage-specific gene sets to infer the location of the given cell population on the trajectory.
#4	Many groups have studied the expression of mammary stem cell and epithelial population-specific genes in bulk breast tumor expression data. If candidate unipotent progenitors play a larger role than previously appreciated (e.g., in LumB or Her2 subsets, Fig. 4a,b), it would be useful to explicitly define the gene sets for candidate unipotent	Relevant new data: Supplementary Fig 17D and Supplementary Table 4 We appreciate an opportunity to explore our data from another aspect. According to this comment, we defined the gene sets for putative progenitor clusters and assessed their associations with breast cancer molecular subtypes. As a result, Lum B and Her2 subtypes had higher LH-pro scores when compared to Lum A subtype. In contrast, Lum A tumors had higher L-Hor scores. We believe that this observation

	progenitors of each epithelial population and reassess their associations with breast cancer molecular subtypes.	added useful indication regarding cells of origins of breast cancer, which have not yet been addressed so far. The gene sets used for this analysis are also explicitly presented in the newly added Supplementary Table 4. We made revisions to the text as follows: (Result, Line 311) When the transcriptome of the human breast cancer was assessed in more detail using the putative progenitor clusters-specific gene sets defined in the mouse epithelial cell data (Supplementary Table 4), LumB and Her2 subtypes had higher LH-pro scores when compared to LumA subtype (Supplementary Fig. 17D). In contrast, LumA tumors had higher L-Hor scores.
--	--	---

		(Discussion, Line 483) In this study, identification of the putative unipotent progenitor populations led to further assessment of cells of origins. The results indicated that LumB type cancers are likely to originate from progenitor cells in the hormone-sensing cell lineage, while LumA tumors would have their origins in more mature L-Hor cells. (Discussion, Line 526) In summary, the integration analysis and identification of putative progenitor populations revealed progenitor cells in the hormone-sensing lineage as putative cells of origin for LumB and Her2 subtypes.
#5	Breast tumors have been subdivided into clinically-distinct groups that extend beyond pam50 subtypes. For example, TNBC can be subdivided into	Relevant new data: Supplementary Fig 18C-E This is another important comment. We assessed the data sets again by considering the suggested classifications (TNBC subtypes, BRCA status, and age).

	Vanderbilt subtypes. Additionally, BRCA status is a key risk factor for TNBC. How do these and other key molecular classifications of breast cancer relate to reference maps defined by the authors? Do the authors observe an association between putative cell-of-origin and age in breast cancer patients?	The findings from additional analyses were mostly confirmatory of the previous literatures, which supported the robustness of the analysis in this study. We added new results (Supplementary Figs 18C-E) and revised the manuscript as follows: (Results, Line 325) The TCGA RNA-seq data were further explored by considering other clinically relevant aspects. Triple-negative breast cancer (TNBC) is a subtype that is characterized by lack of the hormone receptors (ER and PR), combined with the lack of either overexpression or amplification of the HER2 gene⁴². TNBC has been further classified into six (TNBCtype), or more recently four (TNBCtype-4) subtypes by their molecular signatures^{43,44}. When TNBCs in the TCGA datasets were evaluated in light of the lineage genes, most TNBC tumors were mapped onto the Alv lineage
--	---	--

		(Supplementary Fig. 18C). However, the LAR tumors were scattered into the Hor lineage, indicating their different origins in the gland hierarchy. The BRCA gene mutation status contributes to another dimension of heterogeneity in breast cancer. It has been reported that the majority of BRCA1 tumors are basal-like, and BRCA2 tumors are mainly LumB subtype⁴⁵. In accordance with the subtype-lineage relationship in Fig. 3B, BRCA1 tumors were found in the Alv area, while BRCA2 tumors were observed in the both the Alv and Hor lineage with higher Hor scores (Supplementary Fig. 18D). Although age at diagnosis has been also associated with intrinsic subtypes, there is no correlation between age and lineage scores in this cohort (Supplementary Fig. 18E). (Discussion, Line 492) BRCA1 mutation carriers have an expanded luminal
--	--	---

		alveolar population^{6,57}, which would result in the transformation of this cell population later in life. These results supported the robustness of the gene sets-based lineage inference in this study. Interestingly, a part of LAR tumors in TNBC might have different origins in the hormone-sensing lineage. The LAR subtype has been associated with androgen receptor expression and luminal lineage gene signature⁴². Such tumors may lose ER/PR expression during transformation from hormone-sensing cells, while luminal alveolar cells, which are putative cells of origins for most non-LAR TNBCs, usually do not express HRs. Contrary to BRCA1, research on the effects of BRCA2 mutations are currently limited^{58,59}. Future studies should aim at possible dysregulation of L-Hor lineage in BRCA2 mutation carriers.
Minor comments		

Minor #1	Cluster 1 appears to conflate candidate unipotent progenitors of basal epithelial cells with fetal MaSCs. After refining the clustering to separate these two subsets, it will be important to determine whether any cells in the fMaSC cluster are derived from datasets other than Girardi et al. If so, this may reveal evidence for MaSCs later in development. Tracing such cells to their respective datasets and developmental time points would be warranted.	Relevant new data: Supplementary Figs. 8C and 8D Thank you for the comment. We evaluated the cell compositions in each branch (Supplementary Figs. 8C and 8D) by pseudotemporal analysis which separated putative MaSC cells (S5_S3 branch in stream plot) from basal lineage cells (S3_S4). Although putative MaSC cells can be found in pregnant and pubertal glands, their numbers were too low to justify their potential existence. However, we think this information should be shared with readers and have made the following revisions: (Results, Line 166) Putative oligopotent MaSC cells in the S5_S3 branch were not only composed of cells from embryonic glands (Girardi et al.¹⁰), but also from pregnant (Bach et al.¹²) and pubertal glands (Pal et al.¹¹)
-----------------	---	--

		(Supplementary Fig. 8C). When the absolute number of cells was investigated in each dataset, such cells comprised only a very small fraction of the entire dataset. (Discussion, Line 427) The differentiation of MaSC into the luminal lineage was found to occur only in the embryonic gland by the presence of putative bipotent luminal progenitors, indicating that the three different lineages would be maintained by the unipotent progenitors in the adult gland. Putative MaSCs could be present in postnatal glands, but their multipotency would be restricted in normal physiological conditions as indicated in a recent study⁴⁷.
--	--	---

Minor #2	The authors should do a better job in the Results explaining which findings are new and which are confirmatory. The line appears to be blurred in several places	We appreciate the instruction. According to this comment and others, we have significantly revised the entire text to clearly distinguish confirmatory and novel findings and to remove overstatements and irrelevant sentences. As examples of summary sentences, we copied the revised version of the first paragraph of the Discussion and the conclusion below. (Discussion, Line 405) Technical advancement of scRNAseq analysis of the mammary epithelium has led to revisions in our understanding of the biology of the gland, which had largely been investigated by population-level analyses through isolation of distinct, individual cell types. However, the lack of existence of the true stem cell population in a dataset and the inherent differences between scRNAseq studies have limited
-----------------	--	--

		interpretations of the individually collected datasets. Thanks to the recent developments of analytical tools for scRNAseq analyses, our study revealed a putative lineage trajectory that comprehensively covered most of the developmental stages of the mammary gland, which was supported by the five independent studies across three mouse strains, using four different integration algorithms. The integrated data and its reflection to cancer transcriptome comprehensively confirmed the previously suggested differentiation trajectory and cells of origins for human breast cancer, with established catalogues of genes and pathways that are specific to each cell types and species. Our analysis also identified the putative unipotent progenitor populations, which would add important clues to understanding the adult gland homeostasis and breast carcinogenesis. Finally, by referring the scRNAseq data to the lineage trajectory and the
--	--	---

		inferred cells of origin, we visualized how the different developmental stages and the external hormonal exposures can alter the cellular makeup of the mammary epithelium, and ultimately evaluated the risk of the gland for developing specific types of breast cancer. The results from our comprehensive analysis of mouse and human scRNAseq analyses present the mammary epithelium organization and its relationship with breast cancer development in an unprecedented resolution, which could be a good resource in the field. (Discussion, Line 597) In conclusion, we constructed a putative lineage trajectory of the mammary epithelium throughout important WOS by integration of multiple datasets and defined the lineage-specific gene sets to infer the location of the given cell population on the trajectory.
--	--	--

		Our results revisited and added new insights to the relationship between the cellular hierarchy in the gland and the development of the specific subtypes of breast cancer. The catalogue of identified gene/pathway lists and the integrated data are fully accessible in the supplementary data or at the UCSC Cell Browser website (https://mouse-mammary-epithelium-integrated.cells.ucsc.edu), both of which could be a good resource in the mammary gland development and mammary carcinogenesis fields.
--	--	--

Responses to the reviewer 2

Remarks to the author:

In this study the authors report the integration and reanalysis of several (embryonic and adult) mammary epithelial cell (MEC) scRNAseq studies (4 mouse and 1 human). In addition, they also include their own scRNAseq data which investigates the impact of Ovx and hormone treatment on the MEC composition. The authors then use their new integrated data to investigate the cell of origin of breast tumours using publicly available RNAseq data. Overall the authors have done a good job in combining these publicly available datasets which I believe should be made public. However, there several parts of the manuscript where (in my opinion) the authors are over-interpreting the results. If the authors could address these issues I would support publication.

Authors' response:

We appreciate this reviewer's careful evaluation and valuable comments/suggestions. According to this and the other reviewers' comments, we have made significant revisions to our manuscript, including different data integration methods, additional analyses on the progenitor populations, comparisons between data from mouse and human studies, and a data deposit at the UCSC Cell Browser to allow readers to explore genes of interest

interactively. We also removed overinterpretations and overstated sentences to balance our findings and those from others. We hope that our responses properly address the reviewer's comments.

Point-by-point response

#	Reviewer's comment	Authors' response
Major comment		
#1	There are other data integration algorithms (and subsequent interpretation) other than Seurat v3. It would be good to see how the data looks like when other methods are applied. I suggest a comparison in a supplementary figure should be included at least for the overall shape of the data (eg. UMAP).	Relevant new data: Supplementary Figs. 7E-G We appreciate this important comment. We agree that other integration methods should be applied as well. Thus, three additional integration algorithms were applied. While Seurat v3 and Harmony are both anchor-based algorithms, LIGER is an algorithm that exploits both graph- and anchor-based approaches. Meanwhile, scAlign is a deep learning approach based on a neural network. Despite their algorithmic differences, these three additional integration algorithms yielded similar UMAP projections, which substantially strengthens our conclusions in this paper. We have made the data from the four different integration algorithms fully available

to readers and interactively explorable at the UCSC Cell Browser, which is detailed in our response below to comment #10. Accordingly, we revised the manuscript as follows:

(Results, Line 142)

To evaluate the robustness of the results, three additional algorithms (*Harmony*²⁹, *LIGER*³⁰, and *scAlign*³¹) were applied to integrate the five datasets.

The resulting UMAP plots were similar to those obtained by *Seurat v3*: embryonic cells located at the center bridging the three lineages (Supplementary Figs. 7E-G). The integrated data was deposited to the UCSC Cell Browser and interactively explorable on the website (<https://mouse-mammary-epithelium-integrated.cells.ucsc.edu>)³².

(Discussion, Line 410)

		Thanks to the recent developments of analytical tools for scRNAseq analyses, our study revealed a putative lineage trajectory that comprehensively covered most of the developmental stages of the mammary gland, which was supported by the five independent studies across three mouse strains, using four different integration algorithms.
#2	The quality control carried out by the authors excluded low quality cells from all the studies. However, according to this analysis some studies had predominantly low-quality cells (eg. Pal B. et al). This needs to be clearly highlighted in the text and implications of this needs to be mentioned with regard to the data interpretation.	Thank you for the thoughtful comment. We filtered out low quality barcodes and multiplets as carefully and as fair as possible. During our evaluation, we noticed that one sample (Adult_Basal: presorted basal cells from the adult mammary gland) from Pal et al. contained barcodes with much lower feature and UMI counts (Supplementary Fig. 2C). When we integrated six samples in the Pal dataset, that included Adult_Basal, the cells from Adult_Basal failed to merge with other basal cells from adult glands in other samples, possibly due to the lack of marker genes they were supposed to

		have. Therefore, we decided to remove this sample from the analysis. In the original manuscript, this sample only appeared in one figure and was not compared to other samples (PMID: 29158510, Supplementary Figure 5C). However, we realized that analytical methods of scRNAseq is highly critical for readers to evaluate our analyses and they should be clearly highlighted. Accordingly, we revised the manuscript as follows: (Results, Line 90) After reviewing the primary data, one sample (Adult_Basal) from Pal et al. was completely removed from the analysis due to significantly low gene and UMI counts when compared to other samples in the same dataset¹¹ (Supplementary Text and Supplementary Fig. 2C).
--	--	--

		(Discussion, L591) The filtering process removed a considerable number of cells or even a entire sample due to the presence of putative multiplets and low quality cells. This should be carefully interpreted and revisited because analytical pipeline of scRNAseq is still in its infancy.
#3	The 4 mouse datasets were from different genetic backgrounds. This is a key difference and one that needs to be presented clearly. I suggest that the proportions contribution of these genetic backgrounds need to be presented for each cell population/cluster.	Relevant new data: Supplementary Fig. 8C Again, we appreciate the suggestion. In the revised version, we include Supplementary Figure 6B and 8C to show the breakdown of clusters/branches identified in Seurat v3 and Stream, respectively. However, it was hard to comment on potential differences between mouse strains because cells from some important life stages are exclusively from one strain (such as embryonic cells from C57BL/6 and pubertal cells from FVB only). The total number of cells from each strain was also considerably different. Based on these

		assessments, we revised the text as follows: (Results, Line 170) Differences between mouse strains could not be evaluated because cells from some important life stages are exclusively from one strain (such as embryonic cells from C57BL/6 and pubertal cells from FVB only) (Supplementary Figs. 6B and 8C).
#4	Line 160-164. This a big statement regarding the bi-potent luminal progenitors. I think more analysis is needed here. The authors should show the break-down of the data from the 4 different studies and also a breakdown by mouse background.	Relevant new data: Supplementary Figs. 8C and D This is an important comment. Considering comments from both reviewers, we prepared Supplementary Figs. 8C and 8D. Our results suggest that putative MaSC could be in the postnatal glands (especially in pubertal and pregnant glands). However, the bipotent luminal progenitor state was found in embryonic glands almost exclusively. Recently, Centonze et al. reported that putative MaSCs are present in postnatal glands, but

		their multipotency is restricted in normal physiological conditions by luminal cells with TNF signaling after birth (PMID: 32848220). This finding matches our observation that luminal differentiation from MaSC occurs in the embryonic gland only. However, we agree that we cannot guarantee absolutely the presence of “bipotent luminal progenitors” and it should be further evaluated in the future. With additional references, we revised the manuscript as follows: (Result, Line 160) It has been proposed recently that there could be a bipotent luminal progenitor state (S3_S0) in which cell fate would be determined to be a part of of luminal lineage, with the cells being capable of differentiating into either L-Alv and L-Hor cells^{7,34}. However, we have found the predominant occupancy of the S3_S0 branch by the embryonic cells, suggesting that these fate
--	--	---

		determinations occur only during embryonic development. There were only a few, if any, putative bipotent luminal progenitors in the postnatal glands (Supplementary Figs. 8C and D). Putative oligopotent MaSC cells in the S5_S3 branch were not only composed of cells from embryonic glands (Giradi et al.¹⁰), but also from pregnant (Bach et al.¹²) and pubertal glands (Pal et al.¹¹) (Supplementary Fig. 8C). When the absolute number of cells was investigated in each dataset, such cells comprised only a very small fraction of the entire dataset. (Discussion, Line 427, including response to comment #9 of the reviewer 2) The differentiation of MaSC into the luminal lineage was found to occur only in the embryonic gland by the presence of putative bipotent luminal progenitors, indicating that the three different lineages would be
--	--	---

		maintained by the unipotent progenitors in the adult gland. Putative MaSCs could be present in postnatal glands, but their multipotency would be restricted in normal physiological conditions as indicated in a recent study⁴⁷. These results were consistent with the emerging concept of the mammary gland development that have been reported by lineage tracing studies^{2,48,49} and scRNAseq analyses^{3,10,38}. For clarification, it should be noted that different names have been given to the same cell types in the mammary gland. L-Hor cells are analogous to HR+ mature luminal cells (ML) and L-Alv cells correspond to ER-luminal progenitors (LP), or secretory alveolar progenitors, which expand in response to progesterone and during pregnancy and the diestrus phase⁷. There were a couple of scRNAseq studies on the adult mouse mammary gland that reported the presence of the intermediate cell types between L-Alv
--	--	---

		(LP) and L-Hor (ML) clusters, which potentially inferred the presence of bipotent luminal progenitors in adult glands.^{9,11,12}. However, a detailed examination of the data with recently developed algorithms suggested that the cluster was composed of multiplets of the cells from the two luminal clusters. In addition, a luminal intermediate cluster was not found in the other scRNAseq studies of the mammary epithelium^{10,13,14,38}. The lineage tracing studies have also revealed that L-Alv and L-Hor clusters are sustained by the unipotent progenitors in the adult gland⁷. On the other hand, recent scATACseq study supported the presence of bipotent luminal progenitor in adult mammary glands⁵⁰. Therefore, physical validation about the presence of bipotent luminal progenitor in fetal and adult glands will be needed for a definitive conclusion.
#5	There is a major problem with the comparison to the human dataset.	Relevant new data: Supplementary Figs. 15B and 15C

First of all, there is only one study out there with scRNAseq data (there are many more in progress with much larger sample sizes) so it is premature to make any inferences based on this single study. I think the authors should remove the human analysis part and focus the manuscript on the mouse analysis.	Thank you for pointing out this. We understand the risk of making inferences from just one dataset. However, at the same time, our analysis of the TCGA data, based on lineage dataset determined from human scRNAseq, harmonized well with previous literature, which, we believe, supports the robustness of the gene sets. Therefore, we tried to validate our gene sets using different datasets. Although currently there is only one fully published study regarding scRNAseq of human breast epithelium, the analysis included two datasets, one from four individuals with 10X Chromium and the other from three individuals with C1 fluidigm. As we generated gene sets from 10X data (training dataset), we used the data from the Fluidigm as a test dataset. We found that the gene sets clearly separated the test data (from three individuals) on the ternary plot according to their definitive cell types (Supplementary Figs. 15B and C). We understand
--	--

		that both were from one publication, but the applicability to the data from seven individuals across two scRNAseq modalities would support the robustness of the gene sets to some extent. Still, we agreed that we should clearly highlight this limitation. Accordingly, we revised the manuscript as follows: (Results, Line 269) To validate the robustness of the obtained gene sets, scRNAseq data of human breast epithelium from another three individuals sequenced with Fluidigm C1 in a paper of Nguyen et al.³⁸ was analyzed. The data were mounted on Seurat, clustered and definitively annotated according to the original publication (Supplementary Fig. 15B). Then, the data were evaluated by the gene sets obtained from the 10X dataset (Supplementary Fig. 15C). The results showed that the lineage gene sets could clearly indicate
--	--	--

		lineages of cells from the other dataset, which supported the robustness of the method. (Discussion, Line 472) Although the human lineage genes were validated across two different scRNAseq modalities, the data came from only seven individuals in one study. The trajectory of human mammary gland development could be refined further when more relevant human scRNAseq datasets become available. (Discussion, L594) The analysis of human data including the TCGA dataset should be discussed with caution until additional relevant human scRNAseq becomes available and refine the lineage-specific gene sets.
--	--	---

#6	Similarly, the comparison of human scRNAseq to the publicly available cancer datasets is premature in my opinion as the authors are relying on effectively 4 individuals from one study. They should wait till there is more data available.	Thank you for the comment. We agreed with the opinion of the reviewer and revised the manuscript as described above.
#7	It is not clear to me how the Ovx scRNASeq data fits in this paper. This feels like an add on and is also presented in a disjointed way after the human data. I would suggest that the authors either take it out or integrate it into the mouse analysis and discuss it fairly.	We failed to clearly present and explain the meaning of the OVX data among the others in the previous version. One major motivation for this manuscript was to describe the changes in the gland structure and its association with a risk for developing breast cancer during different windows of susceptibility (WOS). Especially in menopausal WOS, hormone replacement therapy and exposure to hormone mimics (endocrine disrupting chemicals, or EDCs) are known to be associated with an increased risk. To our knowledge,

		scRNAseq data for the other major WOS are publically available. However, the extensive analysis of menopausal WOS has been lacking. Therefore, we performed an experiment using the surgically induced model of menopause (OVX) treated with HRT (estrogen and progesterone) and EDC. Considering the reviewer's comment, we moved and integrated the experimental portion before the analysis of the human data (Supplementary Fig. 1). We revised the manuscript as follows: (Introduction, Line 50) Considering the facts that a significant number of breast cancer cases developed in postmenopausal women, and exposure of estrogen or estrogen mimics are thought to promote postmenopausal breast cancer¹⁷, we first designed a new experiment to examine the gland in menopausal WOS and its
--	--	---

		response to external stimuli at a single cell resolution. (Results, Line 66) To reconstruct a complete lineage trajectory of the mammary epithelium, four publically available datasets of the droplet-based scRNAseq of the mouse mammary gland across embryonic, neonatal, pubertal, and pregnant WOS were identified (Fig. 1A, Supplementary Text, and Supplementary Table 1). Furthermore, to address the effect of the loss of ovarian hormones (menopausal WOS), and the impacts of external hormone usage and the environmental exposure to the endocrine disrupting chemicals (EDCs) during that period, we surgically menopausal mice and treated them with 17β-estradiol (E2), progesterone (P4), a mixture of three polybrominated diphenyl ether congeners (PBDEs) [i.e., environmental chemicals interacting with estrogen receptor-alpha (ERα)^{8,9,18}], or
--	--	---

		combinations of them. Image analysis of the whole mammary gland revealed that E2 treatment re-expanded the mammary gland in the surgically menopausal mice with increased total duct length, branching points, and terminal end bud-like structures (TEB-Ls) that are considered to be active proliferation sites of the gland⁹ (Supplementary Fig. 1). The addition of P4, in conjunction with E2, further increased branching of the gland. Simultaneous exposure to PBDEs, potential EDCs, did not have a significant impact on these treatments. However, the PBDE groups did tend to show weaker regrowth of the gland (Supplementary Fig. 1). The mammary glands from these treated mice were analyzed with scRNAseq using the 10x Genomics Chromium v2 single cell 3' RNA-seq platform. (Results, Line 372)
--	--	---

		By knowing that significant structural and functional changes during specific WOS are associated with an increased risk for developing breast cancer, as well as a heightened susceptibility to estrogen, progesterone, and hormone mimics, such as EDCs⁸, we investigated the changes in the mammary gland in different WOS, HRT, and exposure to EDCs in light of the lineage trajectory and inference for the specific types of breast cancer. (Results, Line 390) During the menopausal WOS, the endogenous hormone levels are very low and the mammary tissue is thought to be hyper-sensitive to the exposure of estrogen or its mimics^{9,17}.
#8	The discussion section needs to be toned down significantly. The authors are over interpreting their analysis. This will be made easier if the human data is taken out.	We have significantly revised the entire text to clearly distinguish confirmatory and novel findings and to remove overstatements and irrelevant sentences. As shown above (Comment #5 and 6, reviewer 2), we would like to retain the analysis of the human

		mammary epithelial cell data but with newly added test data. As some examples, we include the revised version of the first paragraph of the Discussion and the Conclusion below. (Discussion, Line 405) Technical advancement of scRNAseq analysis of the mammary epithelium has led to revisions in our understanding of the biology of the gland, which had largely been investigated by population-level analyses through isolation of distinct, individual cell types. However, the lack of existence of the true stem cell population in a dataset and the inherent differences between scRNAseq studies have limited interpretations of the individually collected datasets. Thanks to the recent developments of analytical tools for scRNAseq analyses, our study revealed a putative lineage trajectory that comprehensively covered most of the
--	--	--

		developmental stages of the mammary gland, which was supported by the five independent studies across three mouse strains, using four different integration algorithms. The integrated data and its reflection to cancer transcriptome comprehensively confirmed the previously suggested differentiation trajectory and cells of origins for human breast cancer, with established catalogues of genes and pathways that are specific to each cell types and species. Our analysis also identified the putative unipotent progenitor populations, which would add important clues to understanding the adult gland homeostasis and breast carcinogenesis. Finally, by referring the scRNAseq data to the lineage trajectory and the inferred cells of origin, we visualized how the different developmental stages and the external hormonal exposures can alter the cellular makeup of the mammary epithelium, and ultimately evaluated the risk of the gland for
--	--	--

		developing specific types of breast cancer. The results from our comprehensive analysis of mouse and human scRNAseq analyses present the mammary epithelium organization and its relationship with breast cancer development in an unprecedented resolution, which could be a good resource in the field. (Discussion, Line 597) In conclusion, we constructed a putative lineage trajectory of the mammary epithelium throughout important WOS by integration of multiple datasets and defined the lineage-specific gene sets to infer the location of the given cell population on the trajectory. Our results revisited and added new insights to the relationship between the cellular hierarchy in the gland and the development of the specific subtypes of breast cancer. The catalogue of identified gene/pathway lists and the integrated data are fully accessible in the
--	--	--

		supplementary data or at the UCSC Cell Browser website (https://mouse-mammary-epithelium-integrated.cells.ucsc.edu), both of which could be a good resource in the mammary gland development and mammary carcinogenesis fields.
#9	Line 408-411. Again the authors are making very serious claims by re-interpreting published data. They don't provide any new experimental evidence to support this claim. In addition, the authors fail to mention the scATACseq papers (Wahl lab) that present data which supports the presence of the bi-potent progenitor cells. A more balanced presentation of the data out their (especially if its not their own) is required.	We confirmed that there were adult mammary cells in the branch between the LP/ML and the fetal/basal bifurcations in the Chung et al. paper from Dr. Wahl's lab (PMID: 31597106, Fig. 4A), which indicated the presence of bipotent luminal progenitors. We also realized that the explanation was complicated and confusing because multiple names have been ascribed to the same subset in the mammary gland. Therefore, we revised the manuscript to present ours and others' data in a balanced fashion, with some explanatory sentences to describe the terminologies. (Discussion, Line 427, including response to comment

		#4 of reviewer 2) The differentiation of MaSC into the luminal lineage was found to occur only in the embryonic gland by the presence of putative bipotent luminal progenitors, indicating that the three different lineages would be maintained by the unipotent progenitors in the adult gland. Putative MaSCs could be present in postnatal glands, but their multipotency would be restricted in normal physiological conditions as indicated in a recent study⁴⁷. These results were consistent with the emerging concept of the mammary gland development that have been reported by lineage tracing studies^{2,48,49} and scRNAseq analyses^{3,10,38}. For clarification, it should be noted that different names have been given to the same cell types in the mammary gland. L-Hor cells are analogous to HR+ mature luminal cells (ML) and L-Alv cells correspond to ER-luminal progenitors (LP), or secretory alveolar
--	--	--

		progenitors, which expand in response to progesterone and during pregnancy and the diestrus phase⁷. There were a couple of scRNAseq studies on the adult mouse mammary gland that reported the presence of the intermediate cell types between L-Alv (LP) and L-Hor (ML) clusters, which potentially inferred the presence of bipotent luminal progenitors in adult glands.^{9,11,12}. However, a detailed examination of the data with recently developed algorithms suggested that the cluster was composed of multiplets of the cells from the two luminal clusters. In addition, a luminal intermediate cluster was not found in the other scRNAseq studies of the mammary epithelium^{10,13,14,38}. The lineage tracing studies have also revealed that L-Alv and L-Hor clusters are sustained by the unipotent progenitors in the adult gland⁷. On the other hand, recent scATACseq study supported the presence of bipotent luminal progenitor in adult mammary
--	--	--

		glands⁵⁰. Therefore, physical validation about the presence of bipotent luminal progenitor in fetal and adult glands will be needed for a definitive conclusion.
#10	The combined dataset of the various mouse studies could be a nice resource for the mammary gland community and the authors should consider generating a user friendly website to mine the data.	We greatly appreciate the suggestion. Accordingly, we submitted data to the UCSC Cell Browser (https://mouse-mammary-epithelium-integrated.cells.ucsc.edu) where readers can explore gene expressions of interest, as well as download the processed data. We have provided the link in the manuscript as well (Lines 146, 603, and 764).

REVIEWERS' COMMENTS:

Reviewer #1 (Remarks to the Author):

Overall, the authors have done a very nice job revising the manuscript to address critiques from the prior round of review. That said, there are still a few issues that we would like the authors to address before publication.

1. While we applaud the authors for making an interactive version of the atlas available via the UCSC Cell Browser, we did not see a download link or a file inventory at figshare.com. The authors should ensure that the integrated scRNA-seq atlas is made available for download, including all dataset and cell-level meta-data and annotations, including cluster labels, phenotypes from the original studies, CytoTRACE values, GSVA values, and the coordinates of the integrated embeddings. These data should be made available via figshare.com or GitHub. Separately, the UCSC "Data Download" link stalled and never showed URLs for download.
2. We might have missed it, but gene sets for putative unipotent progenitors should be made available (and clearly labeled) in a supplementary table.
3. Line 66: "To reconstruct a complete lineage trajectory of the mammary epithelium". We understand the authors' ambition, but a "complete" lineage trajectory is clearly an overstatement.
4. Line 485: "The results indicated that LumB type cancers are *likely* to originate from progenitor cells in the hormone-sensing cell lineage". The results support this possibility, but the "likelihood" of this relationship remains unknown. This comment extends to all statements where the authors claim a likely developmental origin based on their results (e.g., line 481). Please temper the wording.

Reviewer #2 (Remarks to the Author):

The authors have addressed all my concerns. The manuscript is much improved. I support its publication.

Responses to the reviewer 1

Remarks to the author:

Overall, the authors have done a very nice job revising the manuscript to address critiques from the prior round of review. That said, there are still a few issues that we would like the authors to address before publication.

Authors' response:

We are very happy to see that our revision answered this reviewer's comments appropriately.

We appreciate the previous comments from this reviewer, which improved the significance, scientific correctness, and readability of our manuscript. We hope that our responses described below will solve the remaining issues.

Point-by-point response

#	Reviewer's comment	Authors' response
#1	While we applaud the authors for making an interactive version of the atlas available via the UCSC Cell Browser, we did not see a download link or a file	Thank you for your careful reviewing comments. According to the editor's recommendation, we redeposited the relevant data and scripts in Zenedo (http://doi.org/10.5281/zenodo.4674274). We also confirmed that "Data Download" tab at the UCSC cell

inventory at figshare.com. The authors should ensure that the integrated scRNA-seq atlas is made available for download, including all dataset and cell-level meta-data and annotations, including cluster labels, phenotypes from the original studies, CytoTRACE values, GSVA values, and the coordinates of the integrated embeddings. These data should be made available via figshare.com or GitHub. Separately, the UCSC "Data Download" link stalled and never showed URLs for download.	browser is currently working and we can download the relevant integrated data. We also copied our Data Availability Statement below for more information. Data availability The authors declare that all data supporting the findings of this study are available within the article, the Supplementary Data, and the data repository or from the corresponding author upon reasonable request. The data from the Tabula Muris Consortium was available in the Figshare with the identifier doi.org/10.1038/s41586-018-0590-4^{13,90}. The other publicly available scRNA datasets were retrieved from the Gene Expression Omnibus under the following accession codes: GSE111113 (https://www.ncbi.nlm.nih.gov/geo/query/acc.cgi?acc=GSE111113, Girradi et al.¹⁰), GSE103275 (https://www.ncbi.nlm.nih.gov/geo/query/acc.cgi?acc
--	---

	=GSE103275, Pal et al.¹¹), GSE106273 (https://www.ncbi.nlm.nih.gov/geo/query/acc.cgi?acc=GSE106273, Bach et al.¹²), GSE113197 (https://www.ncbi.nlm.nih.gov/geo/query/acc.cgi?acc=GSE113197, Human normal breast, Nguyen et al.³⁷), and GSE75688 (https://www.ncbi.nlm.nih.gov/geo/query/acc.cgi?acc=GSE75688, human breast cancer, Chung et al.⁴³). The scRNAseq data obtained in this study were deposited in the Gene Expression Omnibus along with their associated meta data (GSE149949, https://www.ncbi.nlm.nih.gov/geo/query/acc.cgi?acc=GSE149949). The integrated data are explorable on the web browser and can be downloaded as Seurat R objects at https://mouse-mammary-epithelium-integrated.cells.ucsc.edu. The Mouse and human FACS-sorted microarray data of the mammary epithelium were also retrieved from the GSE under the code
--	--

		GSE19446 (https://www.ncbi.nlm.nih.gov/geo/query/acc.cgi?acc=GSE19446) and GSE16997 (https://www.ncbi.nlm.nih.gov/geo/query/acc.cgi?acc=GSE16997), respectively^{7,36}. The TCGA breast cancer data was retrieved from the NCI GDC (https://www.cancer.gov/tcga)³⁹. The data and custom codes in this study were deposited and available in Zenodo (http://doi.org/10.5281/zenodo.4674274)⁶⁹.
#2	We might have missed it, but gene sets for putative unipotent progenitors should be made available (and clearly labeled) in a supplementary table.	Thank you for your comment and instruction. Gene sets for putative unipotent progenitors are provided in Supplementary Data 4. The associated sentence in the main text is copied below. (L267) When the transcriptome of human breast cancer was assessed in more detail using the putative progenitor clusters-specific gene sets defined in the mouse

		epithelial cell data (Supplementary Data 4), LumB and Her2 subtypes had higher LH-pro scores when compared to LumA subtype (Supplementary Figure 17d).
#3	Line 66: "To reconstruct a complete lineage trajectory of the mammary epithelium". We understand the authors' ambition, but a "complete" lineage trajectory is clearly an overstatement.	Thank you again for your careful reviewing and instruction. We agreed with this comment and deleted the indicated and similar sentences. We believe that more balanced expressions are used throughout the revised text.
#4	Line 485: "The results indicated that LumB type cancers are *likely* to originate from progenitor cells in the hormone-sensing cell lineage". The results support this possibility, but the "likelihood"	Thank you for your comments. We also agreed this comment and revised the sentence as follows: (L402) Our results support a possibility that LumB-type cancers are from immature hormone-sensing lineage cells.

of this relationship remains unknown. This comment extends to all statements where the authors claim a likely developmental origin based on their results (e.g., line 481). Please temper the wording.	We also tempered the wording of the similar statements.
--	--

Responses to the reviewer 2

Remarks to the author:

The authors have addressed all my concerns. The manuscript is much improved. I support its publication.

Authors' response:

We appreciate the previous comments from this reviewer, which improved the quality of our manuscript a lot. We are happy to see that the reviewer supports publication of our work.